# A Geometric Analysis of Neural Collapse with Unconstrained Features

**Zhihui Zhu**[*]
University of Denver
zhihui.zhu@du.edu

**Tianyu Ding**
Johns Hopkins University
tding1@jhu.edu

**Jinxin Zhou**
University of Denver
jinxin.zhou@du.edu

**Xiao Li**
University of Michigan
xlxiao@umich.edu

**Chong You**
Google Research
cyou@google.com

**Jeremias Sulam**
Johns Hopkins University
jsulam1@jhu.edu

**Qing Qu**
University of Michigan
qingqu@umich.edu

## Abstract

We provide the first global optimization landscape analysis of *Neural Collapse*—an intriguing empirical phenomenon that arises in the last-layer classifiers and features of neural networks during the terminal phase of training. As recently reported in [1], this phenomenon implies that *(i)* the class means and the last-layer classifiers all collapse to the vertices of a Simplex Equiangular Tight Frame (ETF) up to scaling, and *(ii)* cross-example within-class variability of last-layer activations collapses to zero. We study the problem based on a simplified *unconstrained feature model*, which isolates the topmost layers from the classifier of the neural network. In this context, we show that the classical cross-entropy loss with weight decay has a benign global landscape, in the sense that the only global minimizers are the Simplex ETFs while all other critical points are strict saddles whose Hessian exhibit negative curvature directions. Our analysis of the simplified model not only explains what kind of features are learned in the last layer, but also shows why they can be efficiently optimized, matching the empirical observations in practical deep network architectures. These findings provide important practical implications. As an example, our experiments demonstrate that one may set the feature dimension equal to the number of classes and fix the last-layer classifier to be a Simplex ETF for network training, which reduces memory cost by over 20% on ResNet18 without sacrificing the generalization performance. The source code is available at https://github.com/tding1/Neural-Collapse.

## 1 Introduction

In the past decade, the revival of deep neural networks (DNN) has led to dramatic success in numerous applications ranging from computer vision, to natural language processing, to scientific discovery and beyond [2–5]. Nevertheless, the practice of deep networks has been shrouded with mystery as our theoretical understanding for the success of deep learning remains elusive. There are many intriguing phenomena, such as implicit algorithmic bias in training [6–10], and good generalization of highly-overparameterized networks [7, 11–15], that seem often contradictory to, or cannot be explained by, classical optimization and learning theory.

---

[*]The first two authors contributed to this work equally.

35th Conference on Neural Information Processing Systems (NeurIPS 2021).

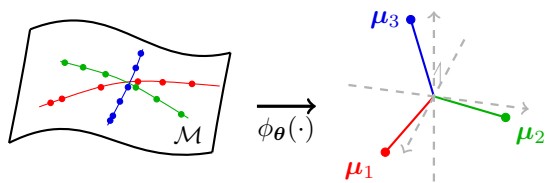

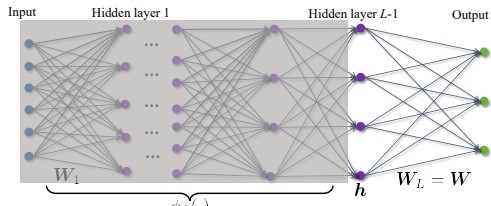

Figure 1: Illustration of Neural Collapse. Here $\phi_{\boldsymbol{\theta}}(\cdot)$ denotes the feature mapping of the network, i.e. the output of the penultimate layer; see (1) for the formal definition.

Figure 2: Illustration of the unconstrained feature model, where the gray box is peeled off so that the representation $\boldsymbol{h}$ is modeled by a simple decision variable for every training sample.

Towards demystifying DNN, recent seminal work [1, 16] empirically discovered an intriguing phenomenon that persists across a range of canonical classification problems during the terminal phase of training. As illustrated in Figure 1, it has been widely observed that last-layer features and classifiers of a trained DNN exhibit simple but elegant mathematical structures:

- **Variability Collapse**: cross-example within-class variability of last-layer features collapses to zero, as the individual features of each class themselves concentrate to their isolated class-means.
- **Convergence to Simplex ETF**: the class-means centered at their global mean are not only linearly separable, but are actually maximally distant and located on a sphere centered at the origin up to scaling (i.e., they form a Simplex Equiangular Tight Frame (ETF) – or Simplex ETF, which is formally defined in Definition C.1 in the Appendix).
- **Convergence to Self-duality**: the last-layer linear classifiers, living in the dual vector space to that of the class-means, are perfectly matched with their class-means.
- **Simple Decision Rule**: the last-layer classifier is behaviorally equivalent to a Nearest Class-Center decision rule.

These results suggest that deep networks are learning maximally separable features between classes, and a max-margin classifier in the last layer upon these learned features, touching the ceiling in terms of the performance. This phenomenon is referred to as *Neural Collapse* ($\mathcal{NC}$) [1], and it persists across a range of canonical classification problems, on different neural network architectures (e.g., VGG [17], ResNet [18], and DenseNet [19]) and on a variety of standard datasets (such as MNIST [20], CIFAR [21], and ImageNet [22]).

Fully demystifying the $\mathcal{NC}$ phenomenon in theory can be very challenging. Perhaps the most difficult hurdle lies in the nonconvexity of the optimization problem for training neural networks, which, loosely speaking, stems from the nonlinear interaction across many different layers of neural networks. Towards this goal, a recent line of work [23–29] studied the properties of last-layer classifiers and features based on the assumption of the so-called *unconstrained feature model* [23] or *layer-peeled model* [26]. At a high level, the unconstrained feature model takes a *top-down* approach to the analysis of deep neural networks [23–26, 29–31], wherein the last-layer features are modeled as *free* optimization variables (hence we call them *unconstrained features*) along with the last-layer classifiers (see Figure 2 for an illustration); this is in contrast to the conventional *bottom-up* approach that studies the problem starting from the input [32–42].[2] The underlying reasoning is that modern deep networks are often highly overparameterized with the capacity of learning any representations [43–46], so that the last-layer features can approximate, or interpolate, any point in the feature space. In this way, the model simplifies the study of last-layer features, enabling us to analyze the interaction between them and the last-layer classifiers.

Nonetheless, the simplified unconstrained feature model still leaves us a highly nonconvex training loss to be dealt with. Despite the nonconvexity, recent work [23–28] studied the global minimizers, proving that Simplex ETFs (i.e., $\mathcal{NC}$) are indeed global solutions to the nonconvex loss. In particular, the work [23, 47] studied the training problem with the least-squared loss, proving that the gradient flow converges to $\mathcal{NC}$ solutions with extra assumptions. Another line of work [24–27] considered the commonly used cross-entropy loss for classification, showing that the only global minimizers of the loss function are Simplex ETFs with different constraints on the weights and features.[3] However,

---

[2]Here, top-down means that we study the network starting from the last-layer down to the input layer, whereas bottom-up refers to an approach from the input up to the last layer.

[3]The constraints on the features are mainly used to prevent it from approaching infinity since the objective function, with the cross-entropy loss, is not coercive. Note that we still refer to this model as an *unconstrained feature model* even if they include norm constraints or regularization.

Table 1: Comparison of the setup and results under the unconstrained feature model with cross-entropy loss.

| | Regularizer | | Bias term | Results | |
| --- | --- | --- | --- | --- | --- |
| | Constraint | Weight decay | | Global minimizer | Landscape |
| [24–27] | ✓ | | | ✓ | |
| This paper | | ✓ | ✓ | ✓ | ✓ |

these results still suffer from several limitations: *(i)* due to the nonconvex nature, only characterizing optimality conditions is not enough to explain the empirical convergence of iterative algorithms to $\mathcal{NC}$, such as stochastic gradient descent (SGD); *(ii)* the problem formulations differ from those typically used in practice, which deploy norm regularization (i.e., weight decay) on the weights, rather than enforcing constraints, for the ease of optimization.[4]

**Contributions of This Work.** Inspired by these pioneering results [1, 23–26, 29], in this work we take a step further by characterizing the global optimization landscape of the network training loss based on the unconstrained feature model. Our contributions are summarized as follows.

- *Benign Global Landscape.* For the unconstrained feature model, we provide the first result showing that a commonly used, regularized cross-entropy loss is a *strict saddle function* [49–51]. In other words, every critical point is either a global solution (corresponding to Simplex ETFs) or a *strict saddle point*[5] with negative curvature, so that there is *no* spurious local minimizer on the optimization landscape. As summarized in Table 1, this is in contrast to previous work [23–26] that only characterizes global minimizers.
- *Efficient, Algorithmic Independent, Global Optimization.* The benign global landscape implies that any method that can escape strict saddle points (e.g. stochastic gradient descent) converges to a global solution [52] that exhibits $\mathcal{NC}$. This result supports our empirical observation, as shown in Section 4.1, that practical overparameterized networks always converge to Simplex ETF solutions with a diverse choice of optimization algorithms.
- *Cost Reduction for Practical Network Training.* Moreover, the universality of $\mathcal{NC}$ implies that there is no need of training the last-layer classifiers since the weights can be simply fixed as a Simplex ETF throughout the training process. On the other hand, since $\mathcal{NC}$ happens whenever $d \geq K$, this implies that we can choose the feature dimension $d$ comparable to the number of classes $K$, reducing the feature dimension for further computational benefits. In Section 4.3, our experiments demonstrate that such a strategy achieves on par performance with classical training methods, leading to substantial cost reductions on both memory and computation.

Our results shed new light on the question raised in the recent paper [53] on the role of the optimization strategy (e.g., stochastic gradient descent) for achieving $\mathcal{NC}$ in training practical deep networks. This question also relates to the recent highly influential work [7] on the implicit algorithmic bias. For multi-class classification problems with linearly separable data, this work [7] showed that linear predictors optimized by gradient descent converge to the max-margin classifiers even without adding any explicit regularization on the cross-entropy loss. Based on this result, a sequence of works [54–61] laid great emphasis on the notion of "inductive bias" of particular optimization algorithms as a reason for the surprising success in training deep learning models.[6] In contrast, both our theoretical result on the global landscape for the unconstrained feature model and the empirical evidence on practical deep models demonstrate that $\mathcal{NC}$ in network training is facilitated *not only* by the choice of the optimization methods, but more importantly, by the choice of loss functions and the power of overparameterization in the network architecture.

## 2 The Problem Setup

A deep neural network is essentially a *nonlinear* mapping $\psi(\cdot) : \mathbb{R}^D \mapsto \mathbb{R}^K$, which can be modeled by a composition of simple maps: $\psi(\boldsymbol{x}) = \psi^L \circ \cdots \circ \psi^2 \circ \psi^1(\boldsymbol{x})$ for $\boldsymbol{x} \in \mathbb{R}^D$, where $\psi^\ell(\cdot)$ $(1 \leq \ell \leq L)$ are called "layers". Each layer is composed of an affine transform, represented by

---

[4]A recent concurrent work [48] studied the gradient flow as well as landscape of the cross-entropy loss under the unconstrained feature model, but without using any constraints or regularizers on the weights.

[5]Throughout the paper, for a minimization problem, we will not distinguish between local maxima and saddle points. We call a critical point *strict saddle* if the Hessian at this point has at least one negative eigenvalue.

[6]While (stochastic) gradient descent and generic steepest descent methods can converge to max-margin classifiers, it should be noted that other commonly used optimization algorithms in deep learning, such as AdaGrad [62] and Adam [63], do not have max-margin properties in general and their solutions depend upon initialization, step-size and other algorithm hyper-parameters [64–66].

some weight matrix $\boldsymbol{W}_\ell$, and bias $\boldsymbol{b}_\ell$, followed by a simple *nonlinear*[7] activation function $\sigma(\cdot)$. More precisely, a vanilla $L$-layer neural network can be written as

$$\psi_{\boldsymbol{\Theta}}(\boldsymbol{x}) \;=\; \boldsymbol{W}_L \underbrace{\sigma\left(\boldsymbol{W}_{L-1} \cdots \sigma\left(\boldsymbol{W}_1 \boldsymbol{x} + \boldsymbol{b}_1\right) + \boldsymbol{b}_{L-1}\right)}_{\phi_{\boldsymbol{\theta}}(\boldsymbol{x})} + \boldsymbol{b}_L. \tag{1}$$

For convenience, we use $\boldsymbol{\Theta} = \{\boldsymbol{W}_k, \boldsymbol{b}_k\}_{k=1}^L$ to denote *all* the network parameters, and use $\boldsymbol{\theta} = \{\boldsymbol{W}_k, \boldsymbol{b}_k\}_{k=1}^{L-1}$ to denote the network parameters up to the last layer. The output of the penultimate layer, denoted by $\phi_{\boldsymbol{\theta}}(\boldsymbol{x})$, is usually referred to as the *representation* or *feature* of the input $\boldsymbol{x}$ learned from the network. In this way, the function implemented by a neural network classifier can also be expressed as a linear classifier acting upon $\phi_{\boldsymbol{\theta}}(\boldsymbol{x})$.

The goal of deep learning is to fit the parameters $\boldsymbol{\Theta}$ so that the output of the model on an input samples $\boldsymbol{x}$ approximates the corresponding output $\boldsymbol{y}$, i.e. so that $\psi_{\boldsymbol{\Theta}}(\boldsymbol{x}) \approx \boldsymbol{y}$, in expectation over a distribution of input-output pairs, $\mathcal{D}$. This can be achieved by optimizing an appropriate loss function $\mathcal{L}(\psi_{\boldsymbol{\Theta}}(\boldsymbol{x}), \boldsymbol{y})$ which quantifies this approximation. In this work, we focus on multi-class classification tasks (say, with $K$ classes), where the class label of a sample $\boldsymbol{x}$ is given by a one-hot vector $\boldsymbol{y} \in \mathbb{R}^K$ representing its membership to one of the $K$ classes. In this setting, cross-entropy is one of the most popular choices for the loss function. Naturally, the distribution $\mathcal{D}$ is unknown, but we have access to training samples that are drawn i.i.d. from $\mathcal{D}$. In this way, one can minimize the empirical risk over these samples by optimizing the following problem

$$\min_{\boldsymbol{\Theta}} \; \sum_{k=1}^K \sum_{i=1}^{n_k} \mathcal{L}_{\mathrm{CE}}\left(\psi_{\boldsymbol{\Theta}}(\boldsymbol{x}_{k,i}), \boldsymbol{y}_k\right) \;+\; \frac{\lambda}{2} \|\boldsymbol{\Theta}\|_F^2, \tag{2}$$

where $\boldsymbol{y}_k \in \mathbb{R}^K$ is a one-hot vector with only the $k$th entry equal to unity ($1 \le k \le K$), $\{n_k\}_{k=1}^K$ are the numbers of training samples in each class, and $\lambda > 0$ is the regularization parameter (or weight decay penalty), and $\mathcal{L}_{\mathrm{CE}}(\cdot, \cdot)$ is the cross-entropy loss. As introduced in Section 1, recent work [1] showed that the features learned by minimizing the above objective showcase the $\mathcal{NC}$ phenomenon: their within-class variability vanishes, and the features converge to a Simplex ETF.

## 2.1 Problem Formulation Based on Unconstrained Feature Models

In deep network models, the nonlinearity and interaction between a large number of layers results in tremendous challenges for analyzing this learning problem. Since modern networks are often highly overparameterized to approximate any continuous function and the characterization of $\mathcal{NC}$ only involves the last-layer features $\phi_{\boldsymbol{\theta}}(\boldsymbol{x})$, a natural idea to simplify the analysis is to treat these features as free optimization variables $\boldsymbol{h} = \phi_{\boldsymbol{\theta}}(\boldsymbol{x}) \in \mathbb{R}^d$, which motivates the name *unconstrained feature model*[8] [23] (see Figure 2 for an illustration). In this way, we can rewrite the network output as $\psi_{\boldsymbol{\Theta}}(\boldsymbol{x}) = \boldsymbol{W}_L \boldsymbol{h} + \boldsymbol{b}_L$.

For simplicity, we consider the setting where the number of training samples in each class is balanced (i.e., $n_k = n$ for all $k \in [K] := \{1, 2, \ldots, K\}$). We also write $\boldsymbol{W} = \boldsymbol{W}_L$ and $\boldsymbol{b} = \boldsymbol{b}_L$ for conciseness. Based on the unconstrained feature model, we consider a slight variant of (2), given by

$$\min_{\boldsymbol{W}, \boldsymbol{H}, \boldsymbol{b}} f(\boldsymbol{W}, \boldsymbol{H}, \boldsymbol{b}) := \frac{1}{Kn} \sum_{k=1}^K \sum_{i=1}^n \mathcal{L}_{\mathrm{CE}}\left(\boldsymbol{W}\boldsymbol{h}_{k,i} + \boldsymbol{b}, \boldsymbol{y}_k\right) + \frac{\lambda_{\boldsymbol{W}}}{2}\|\boldsymbol{W}\|_F^2 + \frac{\lambda_{\boldsymbol{H}}}{2}\|\boldsymbol{H}\|_F^2 + \frac{\lambda_{\boldsymbol{b}}}{2}\|\boldsymbol{b}\|_2^2, \tag{3}$$

with $\boldsymbol{W} \in \mathbb{R}^{K \times d}$, $\boldsymbol{H} = [\boldsymbol{h}_{1,1} \cdots \boldsymbol{h}_{K,n}] \in \mathbb{R}^{d \times N}$ (here, we denote $N = nK$), $\boldsymbol{b} \in \mathbb{R}^K$, and $\lambda_{\boldsymbol{W}}, \lambda_{\boldsymbol{H}}, \lambda_{\boldsymbol{b}} > 0$ are the penalty parameters for the weight decay.

As summarized in Table 1, similar optimization problems have been considered in [24–26]. In contrast to these, our problem formulation here (3), with bias and weight decay, is closer to the loss used in practice for training neural networks; existing work [24–26] considered constrained[9] variants of (3) and without the bias term, which can be implemented but seldom used in practice due to the difficulty of optimization. In the following, we briefly discuss the differences between our simplification and practical settings for training neural networks.

---

[7]The nonlinear operator may include activations such as ReLU [67], pooling, and normalization [68], etc.

[8]This model is also called *layer-peeled model* in [26], where one "peels" off the first $L-1$ layers. It has also been recently studied in [24, 25]. Throughout the paper, we will simply call it unconstrained feature model.

[9]For example, the work [26] considers inequality constraints such that the energy of $\boldsymbol{W}$ and $\boldsymbol{H}$ are bounded; the other work [24, 25] enforces $\boldsymbol{W}$ and $\boldsymbol{H}$ on the spheres up to scaling.

- **Weight Decay on $W$ and $H$.** One simplification of our formulation is in the weight decay. In practice, weight decay is usually imposed on the network parameters $\Theta$, while we enforce weight decay on the last layer's classifier, $W$, and features, $H$. However, this idealization is reasonable since the energy of the features (i.e., $\|H\|_F$) can indeed be upper bounded by the energy of the weights at every layer if the inputs are bounded (which holds in practice), implying that the norm of $H$ is *implicitly* penalized by penalizing $\Theta$. Our experiments in the Appendix demonstrate that both approaches exhibit similar $\mathcal{NC}$ phenomena and comparable performance in practice.
- **Treating the Last-layer Features as Optimization Variables.** One may question that "peeling off" the $L - 1$ layers might oversimplify the problem. Nonetheless, this simplification (which is also adopted in [23–26]) is based on the fact that neural networks with sufficient overparameterization can approximate any function – in Section 4.2, we numerically demonstrate that $\mathcal{NC}$ persists even when we train overparametrized networks on randomly generated labels. Moreover, as we shall see in the following sections, both our theory and experiments demonstrate that our simplification preserves the core properties of last-layer classifiers and features during training – the $\mathcal{NC}$ phenomenon. More specifically, in Section 3 we show that Simplex ETFs are the only global minimizers to our simplified loss function (3), and the loss function is a strict saddle function with no other spurious local minimizers so that it can be optimized efficiently to global optimality.

## 3   Main Theoretical Results

In this section, we present our study on global optimality conditions as well as the optimization landscape of the nonconvex loss in (3).

**Theorem 3.1 (Global Optimality Conditions)** *Assume that the feature dimension $d$ is no smaller than the number of classes $K$, i.e. $d \geq K - 1$, and the number of training samples in each class is balanced, $n = n_1 = \cdots = n_K$. Then any global minimizer $(W^\star, H^\star, b^\star)$ of $f$ in (3) satisfies*

$$w^\star := \left\| \boldsymbol{w}^{\star 1} \right\|_2 = \left\| \boldsymbol{w}^{\star 2} \right\|_2 = \cdots = \left\| \boldsymbol{w}^{\star K} \right\|_2, \quad and \quad \boldsymbol{b}^\star = b^\star \mathbf{1},$$

$$\boldsymbol{h}_{k,i}^\star = \sqrt{\frac{\lambda_W}{\lambda_H n}} \boldsymbol{w}^{\star k}, \quad \forall \, k \in [K], \, i \in [n], \quad and \quad \overline{\boldsymbol{h}}_i^\star := \frac{1}{K} \sum_{j=1}^{K} \boldsymbol{h}_{j,i}^\star = \mathbf{0}, \quad \forall \, i \in [n], \tag{4}$$

*where either $b^\star = 0$ or $\lambda_b = 0$, and the matrix $W^{\star \top} \in \mathbb{R}^{d \times K}$ forms a $K$-Simplex ETF (defined in Definition C.1) up to some scaling, in the sense that the normalized matrix $M := \frac{1}{w^\star} W^{\star \top}$ satisfies*

$$M^\top M = \frac{K}{K-1} \left( I_K - \frac{1}{K} \mathbf{1}_K \mathbf{1}_K^\top \right). \tag{5}$$

At a high level, our proof (in Appendix D) finds lower bounds for the loss in (3) and studies the conditions for the lower bounds to be achieved, similar to [24, 26]. As can be seen in this result, any global solution of the loss function (3) exhibits $\mathcal{NC}$ in the sense that the variability of output features $\{\boldsymbol{h}_{k,i}^\star\}_{i=1}^n$ of each class $k$ ($1 \leq k \leq K$) collapses to zero, and any pair of features $(\boldsymbol{h}_{k_1,i}^\star, \boldsymbol{h}_{k_2,j}^\star)$ from different classes $k_1 \neq k_2$ are maximally separated. Similar results have been obtained in [24–26], which considered different problem formulations, as we have discussed in Section 2.1.

- **Relationship between Class Number $K$ and Feature Dimension $d$.** The requirement that $d \geq K - 1$ is necessary for Theorem 3.1 to hold, simply because $K$ vectors in $\mathbb{R}^d$ cannot form a $K$-Simplex ETF if $K > d + 1$. However, the relationship $d \geq K$ is often true in practice. In general, and in overparameterized models in particular, the feature dimension, $d$, is significantly larger than the number of classes, $K$. For example, the dimension of the features of a ResNet [18] is typically set to $d = 512$ for CIFAR10 [21], a dataset with $K = 10$ classes. This dimension grows to $d = 2048$ for ImageNet [22], a dataset with $K = 1000$ classes.

- **Interpretations on the Bias Term $b^\star$.** In contrast to previous works [24–26], we consider the bias term in the unconstrained feature model (3). Our result indicates that a collapsing phenomenon also exists in the bias term $b^\star$, in the sense that all the elements of $b^\star$ are identical. When the features $H$ are completely unconstrained, our result implies that removing the bias term $b$ has no influence on the performance of the classifier. However, it should be noted that the ReLU unit is often applied at the end of the penultimate layer, so that $H$ should be constrained to be nonnegative, $H \geq \mathbf{0}$. In such cases, $\overline{\boldsymbol{h}}_i^\star$ will no longer be zero, and neither will $b^\star$. Here, the bias

term $\boldsymbol{b}^\star$ will compensate for the global mean of the features, so that the globally-centered features still form a Simplex ETF [1].[10]

## 3.1 Characterizations of the Benign Global Landscape for (3)

The global optimality condition in Theorem 3.1 does not necessarily mean that we can achieve these global solutions efficiently, as the problem is still nonconvex. We now investigate the global optimization landscape of (3) by characterizing all of its critical points. Our next result implies that the training loss is a strict saddle function, and every critical point is either a global minimizer or a strict saddle point that can be escaped using negative curvatures. As a consequence, this implies that the global solutions of the training problem in (3) can be efficiently found from random initializations.

**Theorem 3.2 (No Spurious Local Minima and Strict Saddle Property)** *Assume that the feature dimension is larger than the number of classes, $d > K$, and the number of training samples in each class is balanced $n = n_1 = \cdots = n_K$. Then the function $f(\boldsymbol{W}, \boldsymbol{H}, \boldsymbol{b})$ in (3) is a strict saddle function with no spurious local minimum, in the sense that*

- *Any local minimizer of (3) is a global minimizer of the form shown in Theorem 3.1.*

- *Any critical point $(\boldsymbol{W}, \boldsymbol{H}, \boldsymbol{b})$ of (3) that is not a local minimizer is a strict saddle with negative curvature, i.e. the Hessian $\nabla^2 f(\boldsymbol{W}, \boldsymbol{H}, \boldsymbol{b})$, at this critical point, is non-degenerate and has at least one negative eigenvalue, i.e. $\exists\, i : \lambda_i\left(\nabla^2 f(\boldsymbol{W}, \boldsymbol{H}, \boldsymbol{b})\right) < 0$.*

In a nutshell, our proof relies on connecting the original nonconvex optimization problem (3) to its corresponding low-rank convex counterpart, so that we can obtain the global optimality conditions for (3) based on the latter. With this, we can then characterize the properties of all critical points based on the optimality conditions. We defer all details of this proof to Appendix D.

Existing results [24–26] have *only* studied the global minimizers of the original problem, which has limited implication for optimization. In contrast, Theorem 3.2 characterizes the properties for *all* critical points of the function in (3). As a consequence of this result, many first-order and second-order optimization methods [69] optimizing $(\boldsymbol{W}, \boldsymbol{H}, \boldsymbol{b})$ are guaranteed to converge to a global solution of (3). In particular, the result in [49, 52] ensures that (stochastic) gradient descent with random initialization, the *de facto* optimization algorithm used in deep learning, almost surely escapes strict saddles and converges to a second-order critical point – which happens to be a global minimizer of form showed in Theorem 3.1 for our problem (3).

- **Constructing the Negative Curvature Direction for Strict Saddles.** One of the major difficulties in our proof is to construct the negative curvature direction for strict saddle points. Here, we exploit the fact that the feature dimension $d$ is larger than the number of classes $K$, and construct the negative curvature direction within the null space of $\boldsymbol{W} \in \mathbb{R}^{K \times d}$. This is also the main reason for the requirement $d > K$ in Theorem 3.2, but we conjecture the results also hold for $d = K$ and could be proved with more sophisticated analysis, which is left as future work.

- **Relationship to Low-Rank Matrix Recovery.** As discussed in Appendix A, it has been recently shown that the strict saddle property holds for a wide range of nonconvex problems in machine learning [70–83], including low-rank matrix recovery [78, 80, 84–87]. As we know that $\|\boldsymbol{Z}\|_* = \min_{\boldsymbol{Z} = \boldsymbol{W}\boldsymbol{H}} \frac{1}{2}(\|\boldsymbol{W}\|_F^2 + \|\boldsymbol{H}\|_F^2)$ (see [32] for a proof), our formulation in (3) is closely related to low-rank matrix problems [78, 80, 84–87] with the Burer-Moneirto factorization approach [88], by viewing $\boldsymbol{W}$ and $\boldsymbol{H}$ as two factors of a matrix $\boldsymbol{Z} = \boldsymbol{W}\boldsymbol{H}$. The differences lie in the loss functions and statistical properties of the problem.[11] Thus, our result establishes a connection between the study of low-rank matrix factorization and neural networks under the unconstrained feature model.

---

[10]Suppose that an optimal solution to (3) is $(\boldsymbol{W}^\star, \boldsymbol{H}^\star, \boldsymbol{b}^\star)$, satisfying the conditions in Theorem 3.1. There exists a nonzero vector $\boldsymbol{\alpha} \in \mathbb{R}^d$ such that $\widetilde{\boldsymbol{H}}^\star = \boldsymbol{H}^\star + \boldsymbol{\alpha}\mathbf{1}^\top \geq 0$. Here, $\boldsymbol{\alpha}$ can be viewed as the global mean of $\widetilde{\boldsymbol{H}}^\star$ since $\boldsymbol{H}^\star$ has mean zero. Then, let $\widetilde{\boldsymbol{b}}^\star = -\boldsymbol{W}^\star\boldsymbol{\alpha}$, so that $\widetilde{\boldsymbol{W}}^\star\widetilde{\boldsymbol{H}}^\star + \widetilde{\boldsymbol{b}}^\star\mathbf{1}^\top = \boldsymbol{W}^\star\boldsymbol{H}^\star + (\boldsymbol{W}^\star\boldsymbol{\alpha} + \widetilde{\boldsymbol{b}}^\star)\mathbf{1}^\top = \boldsymbol{W}^\star\boldsymbol{H}^\star$. Therefore, we can see that $(\widetilde{\boldsymbol{W}}^\star, \widetilde{\boldsymbol{H}}^\star, \widetilde{\boldsymbol{b}}^\star)$ achieves the same cross-entropy loss as $(\boldsymbol{W}^\star, \boldsymbol{H}^\star, \boldsymbol{b}^\star)$.

[11]We consider the cross-entropy loss rather than the least-squares loss due to the differences in the task – we focus on classification instead of recovery problems. On the other hand, the results on low-rank matrix recovery are often based on certain statistical properties, such as the randomness in the measurements [84, 85], or restricted well-conditionedness property of the objective function [77, 87]. In contrast, these statistical properties do not exist in our problem, where the model and analysis are purely deterministic.

- **Comparison to Existing Landscape Analysis on Neural Network.** Section 1 provided a comprehensive discussion on the relationship between our result and previous works on landscape analysis for deep neural networks. Although the unconstrained feature model can be viewed as a two-layer linear network with input being the columns of an identity matrix, as preluded in Section 1, our result has much broader implications than the previous results [33, 34, 37, 38, 40, 41, 89]. First, our problem formulation (3) is closer to practical settings for classification tasks, which considers the widely adopted cross-entropy loss while including weight decay and a bias term, while most existing results [33, 34, 37, 38, 40, 41, 89] either do not incorporate any regularization and bias, or focus on the squared loss for the regression problem. More importantly, our result characterizes the precise form of the global solutions (i.e., $\mathcal{NC}$) for the last layer features and classifiers, and shows that they can be efficiently achieved. Moreover, convincing numerical results in [1] and the next section demonstrate that the global solutions do appear and can be achieved by practical networks on various standard image datasets. Our study of last-layer features could have profound implications for studying generalization and robustness of the deep networks.

## 4 Experiments

In this section, we run extensive experiments not only verifying our theoretical results on modern neural networks, but also demonstrating the potential practical benefits of understanding $\mathcal{NC}$. More specifically, while Theorem 3.2 holds true for the simplified unconstrained feature model, in Section 4.1 we run experiments on practical network architectures and show that our analysis of simplified models captures the gist of $\mathcal{NC}$. In particular, we demonstrate that this depends on the geometry of the problem rather than the algorithmic bias, by showing that different types of optimization algorithms *all* achieve $\mathcal{NC}$ during the terminal phase of training. In Section 4.2, we verify the validity of the simplification based on the unconstrained feature model. Moreover, the universality of $\mathcal{NC}$ implies that there is no need for training the last-layer classifiers since the weights can be simply fixed as a Simplex ETF throughout the training process. In Section 4.3, we demonstrate that such a strategy achieves essentially the same generalization performance as classical training algorithms, while improving on memory and computation. We begin by describing the basic experimental setup, including the network architectures, evaluation datasets, training procedures, and metrics for measuring $\mathcal{NC}$.

**Setup of Network Architectures, Dataset, and Training.** In Section 4.1 and Section 4.2, we train a ResNet18 architecture [18] on CIFAR10 [21] for image classification using the cross-entropy loss (2). Due to limited space, we present all the results on MNIST [90] in the Appendix. As is standard, images are normalized (channel-wise) by their mean and standard deviation. We include no data augmentation in this section, as our focus is to study the behavior associated with $\mathcal{NC}$ instead of obtaining state-of-the-art performance. We train the network for 200 epochs with three distinct optimizers: two first-order methods (SGD and Adam) and one second-order method (LBFGS [69]). In particular, we use SGD with momentum 0.9, Adam with $\beta_1 = 0.9, \beta_2 = 0.999$, and LBFGS with a memory size of 10. The initial learning rates for SGD and Adam are set to 0.05 and 0.001, respectively, and decreased by a factor of 10 for every 40 epochs. For LBFGS, we use an initial learning rate of 0.1 and employ a strong Wolfe line-search strategy for subsequent iterations. Except otherwise specified, the weight decay is set to $5 \times 10^{-4}$ for all the experiments.

**Metrics for Measuring $\mathcal{NC}$ During Network Training.** We measure $\mathcal{NC}$ for the learned last-layer classifiers and features based on the properties presented in Section 1. Some of the metrics are similar to those presented in [1]. We first measure the within-class variability collapse by measuring the magnitude of the between-class covariance $\boldsymbol{\Sigma}_B \in \mathbb{R}^{d \times d}$ compared to the within-class covariance $\boldsymbol{\Sigma}_W \in \mathbb{R}^{d \times d}$ of the learned features via $\mathcal{NC}_1 := \frac{1}{K} \operatorname{trace}(\boldsymbol{\Sigma}_W \boldsymbol{\Sigma}_B^{\dagger})$, where $\boldsymbol{\Sigma}_B^{\dagger}$ denotes the pseudo inverse of $\boldsymbol{\Sigma}_B$. For the learned classifier $\boldsymbol{W} \in \mathbb{R}^{K \times d}$, we quantify its closeness to a Simplex ETF up to scaling by $\mathcal{NC}_2 := \left\| \frac{\boldsymbol{W}\boldsymbol{W}^\top}{\|\boldsymbol{W}\boldsymbol{W}^\top\|_F} - \frac{1}{\sqrt{K-1}}\left(\boldsymbol{I}_K - \frac{1}{K}\mathbf{1}_K\mathbf{1}_K^\top\right) \right\|_F$, where we rescale the ETF in (5) so that $\frac{1}{\sqrt{K-1}}(\boldsymbol{I}_K - \frac{1}{K}\mathbf{1}_K\mathbf{1}_K^\top)$ has unit energy (in Frobenius norm). It should be noted that our metric $\mathcal{NC}_2$ combines two metrics used in [1] to quantify to what extent the classifier approaches equiangularity and maximal-angle equiangularity. We then measure the duality between the classifiers $\boldsymbol{W}$ and the centered class-means $\overline{\boldsymbol{H}}$ by $\mathcal{NC}_3 := \left\| \frac{\boldsymbol{W}\overline{\boldsymbol{H}}}{\|\boldsymbol{W}\overline{\boldsymbol{H}}\|_F} - \frac{1}{\sqrt{K-1}}(\boldsymbol{I}_K - \frac{1}{K}\mathbf{1}_K\mathbf{1}_K^\top) \right\|_F$.

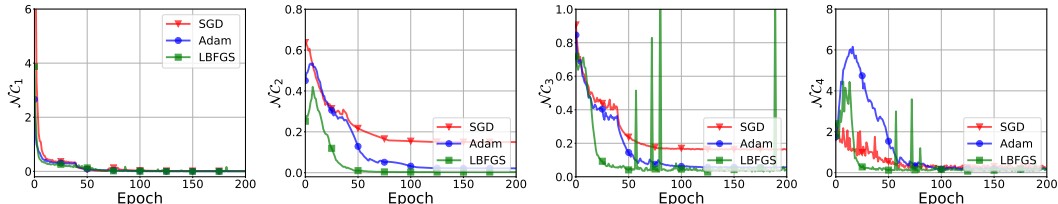

Figure 3: **Illustration of $\mathcal{NC}$ across different training algorithms** with ResNet18 on CIFAR10. From the left to the right, the plots show the four metrics, $\mathcal{NC}_1, \mathcal{NC}_2, \mathcal{NC}_3$, and $\mathcal{NC}_4$, respectively.

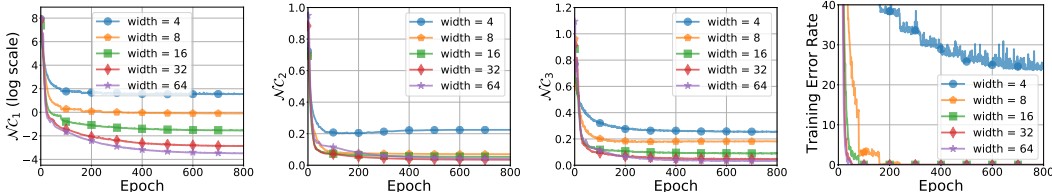

Figure 4: **Training results of ResNet18 with various feature width on CIFAR10 with completely random label.** From the left to the right: $\mathcal{NC}_1, \mathcal{NC}_2, \mathcal{NC}_3$, and the misclassification percentage of training samples.

In many cases, the global mean $\boldsymbol{h}_G$ of the features might not be zero,[12] and the bias term $\boldsymbol{b}$ would compensate for the global mean $\boldsymbol{h}_G$. Thus, we capture this collapsing phenomenon by measuring $\mathcal{NC}_4 := \|\boldsymbol{b} + \boldsymbol{W}\boldsymbol{h}_G\|_2$. The detailed descriptions of the four metrics are given in Appendix B.

## 4.1 The Prevalence of $\mathcal{NC}$ Across Different Optimization Algorithms

We show different types of training methods (e.g., SGD, Adam, and LBFGS) all achieve $\mathcal{NC}$ during the terminal phase of training. Figure 3 shows the evolution of the four metrics $\mathcal{NC}_1, \mathcal{NC}_2, \mathcal{NC}_3$, and $\mathcal{NC}_4$, for measuring $\mathcal{NC}$ as training progresses. We consistently observe that all four metrics collapse to zero, trained by different types of algorithms. This implies that $\mathcal{NC}$ occurs regardless of the choice of training methods. The last-layer features learned by the network are always maximally linearly separable, and correspondingly the last-layer classifier is a perfect linear classifier for the features. See Appendix for the testing performance of the networks learned by different algorithms.

## 4.2 The Validity of (3) Based on Unconstrained Feature Models for $\mathcal{NC}$

The premise of our global landscape analysis of (3) for studying $\mathcal{NC}$ in deep neural networks is based upon the unconstrained feature model introduced in Section 2.1, which simplifies the network by synthesizing the first $L - 1$ layers as a universal approximator that generates a simple decision variable for each training sample. Here, we demonstrate through experiments that such a simplification is reasonable for overparameterized networks, in the sense that they are sufficient for characterizing $\mathcal{NC}$ in practical network training. In particular, we demonstrate that overparameterization is crucial for $\mathcal{NC}$ phenomenon during network training, while the input plays minimal influence. To that goal, we modify the training dataset CIFAR10 by replacing *all* the correct label for each training sample with a *random* counterpart.[13] We report the corresponding $\mathcal{NC}$ behaviors in Figure 4, which shows how training misclassification rate and $\mathcal{NC}$ evolve over epochs of training for networks with different widths[14]. As the network is sufficiently large, it has enough capacity to memorize the training data and achieves zero training error, which is consistent with the observations in [11]. Moreover, we find from Figure 4 that the training accuracy is highly correlated with $\mathcal{NC}$ in the sense that a larger network (i.e., larger width) tends to exhibit severe $\mathcal{NC}$ and achieves smaller training error. In other words, while the emerging consensus is that the network can interpolate any training data, our results show that such interpolation happens in a particular way – the features are maximally separated, followed by a max-margin linear classifier. In Appendix B, we also report experiments on weight decay imposed on the features $\boldsymbol{H}$, as in (3).

---

[12]For example, as discussed after Theorem 3.1 all the feature vectors in $\boldsymbol{H}$ would be nonnegative, because the nonnegative nonlinear operator ReLU has been applied at the end of the penultimate layer.

[13]We also conducted experiments on completely random dataset where each training image is generated with pixels uniformly from $[0, 1]$ and we observed similar results.

[14]Here, for ResNet18 we adopt the method in [15] to change its network width.

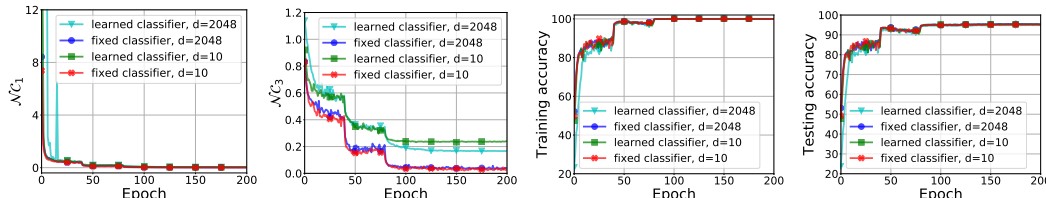

Figure 5: **Comparison of the performances of ResNet 50 with learned vs. fixed last-layer classifiers on CIFAR10.** From left to right): $\mathcal{NC}_1$, $\mathcal{NC}_3$, Training Accuracy, Testing Accuracy.

### 4.3  Insights from $\mathcal{NC}$ for Improving Network Designs

Finally, we conduct exploratory experiments to demonstrate the practical benefits of $\mathcal{NC}$ phenomenon. The universality of $\mathcal{NC}$ implies that the final classifier (i.e. the $L$-th layer) of a neural network always converges to a Simplex ETF, which is fully determined up to an arbitrary rotation and happens when $K \le d$. Thus, based on these understandings, we show that we can substantially improve the computational cost by modifying the architecture without the sacrificing performance, by *(i)* fixing the last-layer classifier as a Simplex ETF[15], and *(ii)* reducing the feature dimension $d = K$. Here, to demonstrate our method can achieve state-of-the-art performance, we do include data augmentation in the training of our ResNet50 model [91] on CIFAR10, achieving around 95% test accuracy. See Appendix B for the results on MNIST and CIFAR10 with ResNet18.

**Fixing the Last-layer Classifier as a Simplex ETF.**   Figure 5 presents a comparison of learned and fixed classifiers in terms of within-class variation collapse ($\mathcal{NC}_1$), self-duality ($\mathcal{NC}_3$), training accuracy, and test accuracy. These results imply that the fixed classifier exhibits the same within-class variation collapse for the features $\boldsymbol{H}$, and achieves the same classification accuracy as the *fully-trained* classifier. On the other hand, fixing the classifier can reduce the number of parameters and the computational complexity for training. The number of parameters in the classifier can be significant for tasks with a large number of classes and large feature dimensions. For example, for ImageNet, a dataset with $K = 1000$ classes, fixing the classifier can reduce 8.01%, 11.76%, and 52.56% of total learning parameters for ResNet50, DenseNet169 [19], and ShuffleNet [92], respectively. We note that our result also provides a theoretical justification for the work in [93] that fixes the classifier as orthonormal matrices. Indeed, these are close to simplex ETFs, particularly when the number of classes is large.

**Feature Dimension Reduction for $\boldsymbol{H} \in \mathbb{R}^{d \times nK}$ by Choosing $d = K$.**[16]   In many classification problems, the practice of deep learning typically uses a feature dimension $d$ that is much larger than the number of classes $K$. In contrast, $\mathcal{NC}$ implies that there is no need to choose a $d$ that is much larger than the number of classes $K$. Reducing the dimension $d$ can lead to substantial reductions in memory and computation cost. As shown in Figure 5, we also train all the weights of ResNet50 on CIFAR10 using SGD with $d = K$. The results demonstrate that $\mathcal{NC}$ persists even when we choose $d = K$, and the network achieves on-par performance with networks of large $d$, in terms of training and test accuracy. This implies that when the number of classes $K$ is small, we can choose a small feature dimension $d = K$ (or $d \gtrsim K$) instead of using a large universal $d$ to reduce the computation and memory costs for training. By setting $d = K$, this reduces the amount of parameters and hence the memory cost in ResNet18 and ResNet50 by 20.70% and 4.45% respectively.

## 5  Conclusion

In this work, we have provided an in-depth analysis to demystify the $\mathcal{NC}$ phenomenon, which appears during the terminal phase of training deep networks in classification problems. Based on the unconstrained feature model [24–26], we proved that Simplex ETFs are the only global minimizers of the cross-entropy training loss with weight decay and bias. Moreover, we showed that the loss function is a strict saddle function with respect to the last-layer features and classifiers, with no other spurious local minimizers. In contrast to existing landscape analyses for deep neural networks, which mostly focus on the optimization perspective, our simplified analysis not only characterizes

---

[15]Specifically, we set $\boldsymbol{W}^{\top} = \sqrt{\frac{K}{K-1}} \boldsymbol{P} \left( \boldsymbol{I}_K - \frac{1}{K} \mathbf{1}_K \mathbf{1}_K^{\top} \right)$ where $\boldsymbol{P} \in \mathbb{R}^{d \times K}$ contains the first $K$ columns of a $d \times d$ identity matrix, which lifts a $K \times K$ ETF to $d \times K$ matrix. For simplicity, we also learn the bias term in the last layer, though our result indicates it can be set as $\boldsymbol{W}\overline{\boldsymbol{h}}$, where $\overline{\boldsymbol{h}}$ is the global mean of the features.

[16]Though Theorem 3.2 requires $d > K$, we conjecture it also holds for $d = K$ as discussed in Section 3.1.

the features that are learned in the last layer, but also explains why they can be efficiently optimized. This provides support for empirical observations in practical deep network architectures. Moreover, the study of last-layer features could have profound implications for optimization, generalization, and robustness of broad interests, which are the subjects of future work. It is also of interest to extend the current study to the case where $d < K$, which is the case in contrastive learning [94, 95] and many applications, such as recommendation systems [96] and document retrieval [97].

## Acknowledgment

ZZ acknowledges support from NSF grant CCF 2008460. XL and QQ acknowledge support from NSF grant DMS 2009752. JS acknowledges support from NSF grant CCF 2007649. We would like to thank Qinqing Zheng (Facebook AI Research), Vardan Papyan (U. Toronto), and Felix Yu (Google Research) for timely pointing us to some important references and valuable feedback on the final draft. We thank Christina Baek (UC Berkeley) and Sam Buchanan (Columbia U.) for fruitful discussions during various stages of the work. We also thank Zhexin Wu (U. Michigan) for proofreading and pointing out several typos in the draft, and the four anonymous reviewers for their constructive comments.

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
