# Appendices

**Notations and Organizations.** To begin, we first briefly introduce some notations used throughout the appendix. For a scalar function $f(\boldsymbol{Z})$ with a variable $\boldsymbol{Z} \in \mathbb{R}^{K \times N}$, its gradient is a $K \times N$ matrix whose $(i,j)$-th entry is $[\nabla f(\boldsymbol{Z})]_{ij} = \frac{\partial f(\boldsymbol{Z})}{z_{ij}}$ for all $i \in [K], j \in [N]$, where $z_{ij}$ represents the $(i,j)$-th entry of $\boldsymbol{Z}$. The Hessian of $f(\boldsymbol{Z})$ can be viewed as an $KN \times KN$ matrix by vectorizing the matrix $\boldsymbol{Z}$. An alternative way to present the Hessian is by a bilinear form defined via $[\nabla^2 f(\boldsymbol{Z})](\boldsymbol{A}, \boldsymbol{B}) = \sum_{i,j,k,\ell} \frac{\partial^2 f(\boldsymbol{Z})}{\partial z_{ij} z_{k\ell}} a_{ij} b_{k\ell}$ for any $\boldsymbol{A}, \boldsymbol{B} \in \mathbb{R}^{K \times N}$, which avoids the procedure of vectorizing the variable $\boldsymbol{Z}$. We will use the bilinear form for the Hessian in Appendix E.

The appendix is organized as follows. In Appendix A, we discuss the relationship between this work to the previous work that beyond neural collapse. Appendix B includes the detailed description of metrics for measuring $\mathcal{NC}$ during network training and additional experimental results. In Appendix C, we introduce the basic definitions and inequalities used throughout the appendices. In Appendix D, we provide a detailed proof for Theorem 3.1, showing that the Simplex ETFs are the *only* global minimizers to our regularized cross-entropy loss. Finally, in Appendix E, we present the whole proof for Theorem 3.2 that the function is a strict saddle function and no spurious local minimizers exist, which is one of the major contributions of the work.

## A    Relationship to the Prior Arts Beyond Neural Collapse

Our work highly relates to recent advances on studying the optimization landscape in neural network training; see [98, 99] for a contemporary survey. Most of the existing work [32–42] analyzes the problem based on a *bottom-up* approach – from the input to the output of neural networks – ranging from two-layer linear network [33, 38, 40, 89], deep linear network [34, 37, 38, 41], to nonlinear network [35,36,38,39]. More specifically, the line of work [33,34,38,40,41] studied the optimization landscape for linear two-layer networks and proved that the associated training loss is a strict saddle function. For deeper linear networks, it can be shown that flat saddle points exist at the origin, but there are no spurious local minima [34, 37]. For nonlinear neural networks, it has been proved that there do exist spurious local minima [35, 36], but such local minima may be eliminated, or the number can be significantly reduced, in the over-parameterization regime [35, 39]. Additionally, the work [32] proved that certain local minima (having an all-zero "slice") are also global solutions, but the analysis is crucially dependent on the sufficient condition of an all-zero slice in the weights, which is insufficient to characterize the landscape properties. At a high level, the differences between these results and ours can be summarized as follows.

- *A Feature Learning Perspective.* While most of these results based on the bottom-up approach explain optimization and generalization of certain types of deep neural networks, they provided limited insights into the practice of deep learning. In contrast, we take a *top-down* approach to look at the network starting from the very last layer. The slight difference in the models can lead to a dramatic difference in the interpretability for deep learning. By starting from the last layer, our results not only provide valid reasons on why the training loss can be efficiently optimized, but also provide a precise characterization of the last-layer features as well as the classifiers learned from the network. As we will show, such a feature learning perspective not only helps with network design (see Section 4.3), but may bear broadly on generalization and robustness of deep learning as well as the recent development of contrastive learning (see Section 5).

- *Connections to Empirical Phenomena.* Moreover, most existing theoretical results on landscape analysis [100, 101] are somewhat disconnected from practice due to unrealistic assumptions, providing limited guiding principles for modern network training or design. In contrast, our assumption on the last-layer features as optimization variables is naturally based on model overparameterization. Moreover, our results provide explanations for $\mathcal{NC}$, an empirical phenomenon that has been widely observed on convincing numerical evidence across many different practical network architectures and a variety of standard image datasets.

Our work also broadly relates to the recent theoretical study for deep learning based on Neural Tangent Kernels (NTKs) [102], where a neural network behaves like a *linear* model on random features hence has a benign optimization landscape. However, the "kernel regime" that NTKs work

in requires neural networks that are infinitely wide – or at least so wide that is beyond the regime that practical neural networks work in [103–106]. In contrast, we adopt the unconstrained feature model which does not directly impose requirements on the width of the neural network and, as shown in our experiments, well captures the behavior of practical neural networks. Hence, our result can provide a more practical understanding for deep learning.

Moreover, from a boarder perspective our work is rooted in recent advances on global nonconvex optimization theory for signal processing and machine learning problems [51, 80, 82, 107, 108]. In a sequence of works [71–83, 109–111], the authors showed that many problems exhibit "equivalently good" global minimizers due to symmetries and intrinsic low-dimensional structures, and the loss functions are usually strict saddles [49–51]. These problems include, but are not limited to, phase retrieval [75, 76], low-rank matrix recovery [71, 74, 77, 79, 80], dictionary learning [70, 72, 73, 81, 82], and sparse blind deconvolution [83, 109–111]. As we shall see, the global minimizers (i.e., simplex ETFs) of our problem here also exhibit a similar rotational symmetry, compared to low-rank matrix recovery. In fact, our proof techniques are inspired by recent results on low-rank matrix recovery [77, 80]. Thus, our work establishes a new connection between recent advances on nonconvex optimization theory and deep learning.

## B    Experiments

In this section, we provide with more details for reproducing the experiments presented in the paper. In particular, we emphasize that the datasets involved in the paper, namely MNIST and CIFAR10, are publicly available for academic usage: MNIST dataset is made available under the terms of the Creative Commons Attribution-Share Alike 3.0 license, and CIFAR10 dataset is made available under the MIT license. All experiments are conducted on a single RTX3090 GPU with 24GB memory.

### B.1    Metrics for Measuring $\mathcal{NC}$ During Network Training.

We measure $\mathcal{NC}$ for the learned last-layer classifiers and features based on the properties presented in Section 1. Some of the metrics are similar to those presented in [1]. First, we define the global mean and class mean of the last-layer features $\{\boldsymbol{h}_{k,i}\}$ as

$$\boldsymbol{h}_G \;=\; \frac{1}{nK} \sum_{k=1}^K \sum_{i=1}^n \boldsymbol{h}_{k,i}, \quad \overline{\boldsymbol{h}}_k \;=\; \frac{1}{n} \sum_{i=1}^n \boldsymbol{h}_{k,i} \; (1 \le k \le K).$$

- **Within-class Variability Collapse for the Learned Features $\boldsymbol{H}$.** We introduce the within-class and between-class covariance matrices as

$$\boldsymbol{\Sigma}_W \;:=\; \frac{1}{nK} \sum_{k=1}^K \sum_{i=1}^n \left(\boldsymbol{h}_{k,i} - \overline{\boldsymbol{h}}_k\right)\left(\boldsymbol{h}_{k,i} - \overline{\boldsymbol{h}}_k\right)^\top, \quad \boldsymbol{\Sigma}_B \;:=\; \frac{1}{K} \sum_{k=1}^K \left(\overline{\boldsymbol{h}}_k - \boldsymbol{h}_G\right)\left(\overline{\boldsymbol{h}}_k - \boldsymbol{h}_G\right)^\top.$$

Thus, we can measure the within-class variability collapse by measuring the magnitude of the between-class covariance $\boldsymbol{\Sigma}_B \in \mathbb{R}^{d \times d}$ compared to the within-class covariance $\boldsymbol{\Sigma}_W \in \mathbb{R}^{d \times d}$ of the learned features via

$$\mathcal{NC}_1 \;:=\; \frac{1}{K} \operatorname{trace}\left(\boldsymbol{\Sigma}_W \boldsymbol{\Sigma}_B^\dagger\right), \tag{6}$$

where $\boldsymbol{\Sigma}_B^\dagger$ denotes the pseudo inverse of $\boldsymbol{\Sigma}_B$.

- **Convergence of the Learned Classifier $\boldsymbol{W}$ to a Simplex ETF.** For the learned classifier $\boldsymbol{W} \in \mathbb{R}^{K \times d}$, we quantify its closeness to a Simplex ETF up to scaling by

$$\mathcal{NC}_2 \;:=\; \left\| \frac{\boldsymbol{W}\boldsymbol{W}^\top}{\|\boldsymbol{W}\boldsymbol{W}^\top\|_F} - \frac{1}{\sqrt{K-1}} \left(\boldsymbol{I}_K - \frac{1}{K}\mathbf{1}_K\mathbf{1}_K^\top\right) \right\|_F, \tag{7}$$

where we rescale the ETF in (5) so that $\frac{1}{\sqrt{K-1}}\left(\boldsymbol{I}_K - \frac{1}{K}\mathbf{1}_K\mathbf{1}_K^\top\right)$ has unit energy (in Frobenius norm). It should be noted that our metric $\mathcal{NC}_2$ combines two metrics used in [1] to quantity to what extent the classifier approaches equiangularity and maximal-angle equiangularity.

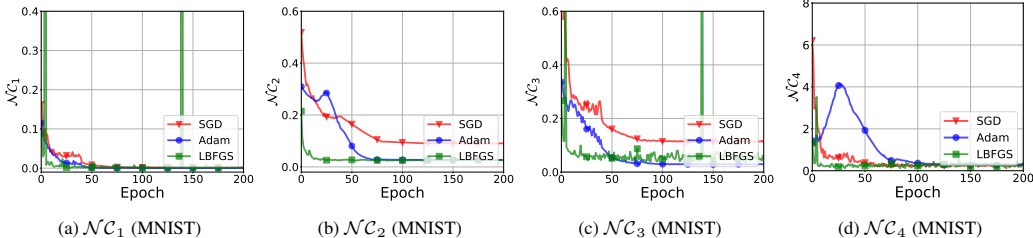

(a) $\mathcal{NC}_1$ (MNIST)  (b) $\mathcal{NC}_2$ (MNIST)  (c) $\mathcal{NC}_3$ (MNIST)  (d) $\mathcal{NC}_4$ (MNIST)

Figure 6: **Illustration of $\mathcal{NC}$ across different training algorithms** with ResNet18 on MNIST. From the left to the right, the plots show the four metrics, $\mathcal{NC}_1, \mathcal{NC}_2, \mathcal{NC}_3,$ and $\mathcal{NC}_4$, for measuring $\mathcal{NC}$, defined in (6), (7), (8), and (9), respectively.

- *Convergence to Self-duality.* Next, we measure the collapse of the learned features $H$ to its dual $W$. Let us define the centered class-mean matrix as

$$\overline{H} := \begin{bmatrix} \overline{h}_1 - h_G & \cdots & \overline{h}_K - h_G \end{bmatrix} \in \mathbb{R}^{d \times K}.$$

Thus, we measure the duality between the classifiers $W$ and the centered class-means $\overline{H}$ by

$$\mathcal{NC}_3 := \left\| \frac{W\overline{H}}{\|W\overline{H}\|_F} - \frac{1}{\sqrt{K-1}} \left( I_K - \frac{1}{K} \mathbf{1}_K \mathbf{1}_K^\top \right) \right\|_F. \tag{8}$$

- *Collapse of the Bias.* In many cases, the global mean $h_G$ of the features might not be zero,[17] and the bias term $b$ would compensate for the global mean $h_G$ in the sense that

$$W h_{k,i} + b = W(h_{k,i} - h_G) + \underbrace{W h_G + b}_{=\mathbf{0}}.$$

Thus, we capture this collapsing phenomenon by measuring

$$\mathcal{NC}_4 := \|b + W h_G\|_2. \tag{9}$$

## B.2  Additional Experiments

### B.2.1  The Prevalence of $\mathcal{NC}$ Across Different Optimization Algorithms

In Figure 6 and Figure 7, we run all the experiments with ResNet18 on MNIST and CIFAR10 without modification. The results lead to the following observations:

- $\mathcal{NC}$ *is Algorithm Independent.* Similar to Figure 3, we consistently observe from Figure 6 that all four metrics collapse to zero, trained by different types of algorithms – SGD, Adam, and LBFGS. This implies that $\mathcal{NC}$ happens regardless of the choice of training methods. The last-layer features learned by the network are always maximally linearly separable, and correspondingly the last-layer classifier is a perfect linear classifier for the features.

- *Relationship between $\mathcal{NC}$ and Generalization.* Figure 7 depicts the learning curves in terms of both the training and test accuracy for all three optimization algorithms (i.e., SGD, Adam, and LBFGS). These experimental results[18] show that different training algorithms learn neural networks with notably different generalization performances, even though all of them exhibit $\mathcal{NC}$. Since $\mathcal{NC}$ is only a characterization of the training data, it does not directly translate to unseen data. As the network is highly overparameterized, there are infinitely many networks that produce the same $H$ with $\mathcal{NC}$ for a particular training dataset, but with different generalization performance. This suggests that study generalization needs to consider the algorithmic bias and the learned weights for the feature $H$. A thorough investigation between $\mathcal{NC}$ and generalization is the subject of future work.

---

[17]For example, as discussed after Theorem 3.1 all the feature vectors in $H$ would be nonnegative, because the nonnegative nonlinear operator ReLU has been applied at the end of the penultimate layer.

[18]Note that here we use the default version of LBFGS in PyTorch. Other variants of quasi-Newton methods [112, 113] may give different or better generalization performance.

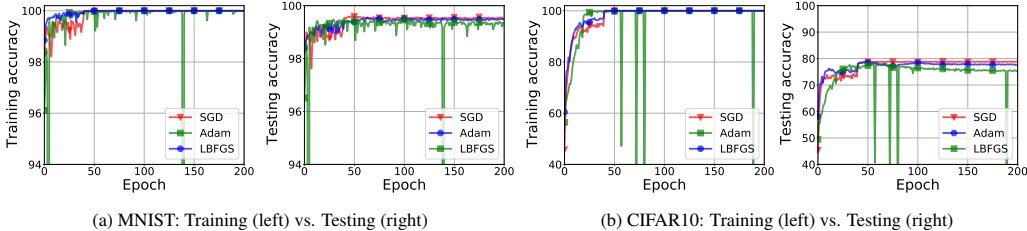

(a) MNIST: Training (left) vs. Testing (right)    (b) CIFAR10: Training (left) vs. Testing (right)

Figure 7: **Illustrations of training and test accuracy** for three different training algorithms (i.e., SGD, Adam, and LBFGS) with ResNet18 on MNIST and CIFAR10.

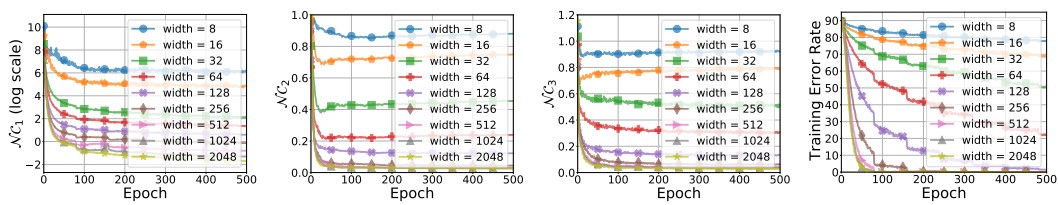

(a) Random Label CIFAR10-MLP (from left to right): $\mathcal{NC}_1$ (log scale), $\mathcal{NC}_2$, $\mathcal{NC}_3$, Training Error Rate

Figure 8: **Training results of CIFAR10 with completely random label.** Multilayer perceptron (MLP) of a fixed depth of 4 is used with various feature width. Note that the right column is the misclassification percentage of all samples in training.

### B.2.2    The Validity of (3) Based on Unconstrained Feature Models for $\mathcal{NC}$

The premise of our global landscape analysis of (3) for studying $\mathcal{NC}$ in deep neural networks is based upon the unconstrained feature model introduced in Section 2.1, which simplifies the network by synthesizing the first $L-1$ layers as a universal approximator that generates a simple decision variable for each training sample. Here, we demonstrate through experiments that such a simplification is reasonable for overparameterized networks, at least sufficient for characterizing $\mathcal{NC}$ in practical network training.

**The Validity of Unconstrained Feature Models.**    For the ease of studying the effects of model sizes (i.e., overparameterization) on $\mathcal{NC}$, we also train 4-layer multilayer perceptrons (MLP) of different network width using SGD with learning rate $0.01$ and weight decay $10^{-4}$. We report the corresponding $\mathcal{NC}$ behaviors in Figure 8, which shows how training misclassification rate and $\mathcal{NC}$ evolve over epochs of training for networks with different widths. Figure 8 shows similar results to Figure 4 with ResNet18, in the sense that the training accuracy is highly correlated with $\mathcal{NC}$ in the sense that a larger network (i.e., larger width) tends to exhibit severe $\mathcal{NC}$ and achieves smaller training error.

**Comparison of Weight Decay on the Network Parameter $\Theta$ vs. on the Features $H$ in (3).** In comparison to typical training protocols of deep networks which enforce weight decay on all the network weights $\Theta$, our problem formulation (3) based on the unconstrained feature model replaced $\Theta$ by penalizing the feature $H$ produced by the $L-1$ "peeled-off" layers. To check the practicality of such formulation, we empirically run experiments using ResNet18 on MNIST and CIFAR10. Figure 9 shows the $\mathcal{NC}$ evolution for both the classical formulation and our "peeled" formulation, we notice that the $\mathcal{NC}$ behavior happens in both scenarios comparably. We also point that without extensive hyper-parameter tuning, the models trained under the "peeled" set-up could already achieve test accuracy of $99.57\%$ and $77.92\%$ on MNIST and CIFAR10 respectively. We note that such performances are on-par with the test accuracy of the classical formulation (2), with test accuracy of $99.60\%$ and $78.42\%$ on MNIST and CIFAR10 as reported in Figure 7.

### B.3    Insights from $\mathcal{NC}$ for Improving Network Designs

Similar to Figure 5, we train a ResNet18 and fix the weights in the last layer as a Simplex ETF throughout training for MNIST and CIFAR10 datasets, but without any data augmentation. Fig-

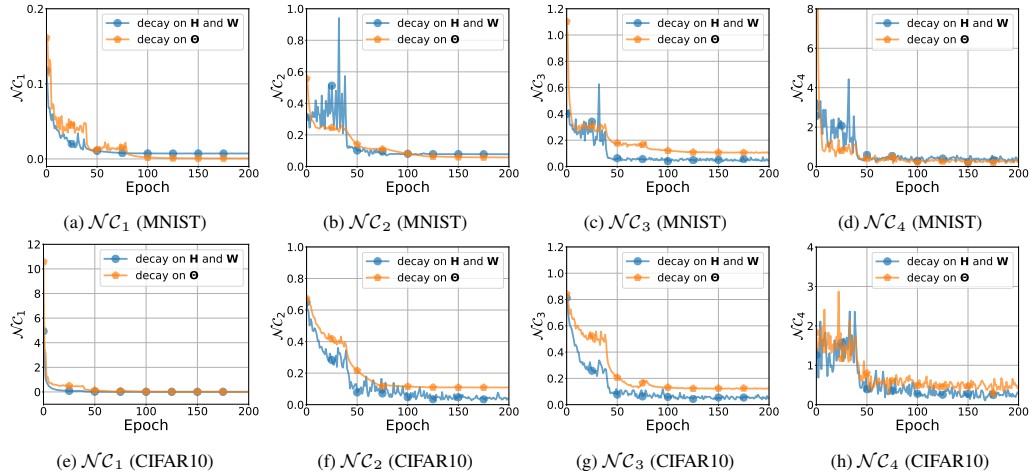

(a) $\mathcal{NC}_1$ (MNIST)     (b) $\mathcal{NC}_2$ (MNIST)     (c) $\mathcal{NC}_3$ (MNIST)     (d) $\mathcal{NC}_4$ (MNIST)

(e) $\mathcal{NC}_1$ (CIFAR10)     (f) $\mathcal{NC}_2$ (CIFAR10)     (g) $\mathcal{NC}_3$ (CIFAR10)     (h) $\mathcal{NC}_4$ (CIFAR10)

Figure 9: **Comparison of $\mathcal{NC}$ behavior for weight decay on $\Theta$ vs. on $(\boldsymbol{H}, \boldsymbol{W})$.** For the latter set up, we choose $\lambda_{\boldsymbol{W}} = \lambda_{\boldsymbol{b}} = 0.01$ and $\lambda_{\boldsymbol{H}} = 0.00001$.

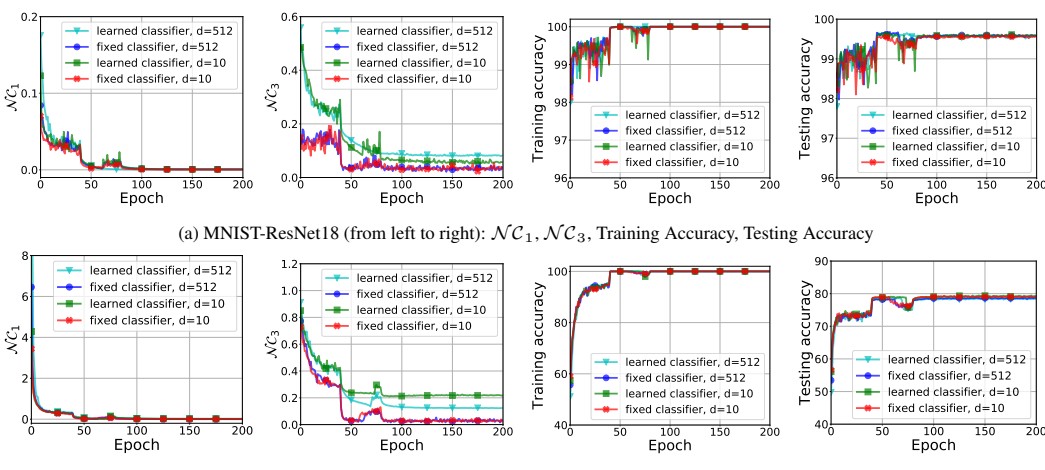

(a) MNIST-ResNet18 (from left to right): $\mathcal{NC}_1, \mathcal{NC}_3$, Training Accuracy, Testing Accuracy

(b) CIFAR10-ResNet18 (from left to right): $\mathcal{NC}_1, \mathcal{NC}_3$, Training Accuracy, Testing Accuracy

Figure 10: **Comparison of the performances on learned vs. fixed last-layer classifiers.** We compare within-class variation collapse $\mathcal{NC}_1$, self-duality $\mathcal{NC}_3$, training accuracy, and test accuracy, on fixed and learned classifier on MNIST-ResNet18 (Top) and CIFAR10-ResNet18 (Middle). All networks are trained by SGD optimizer.

ure 10 presents a comparison of learned and fixed classifiers in terms of within-class variation collapse ($\mathcal{NC}_1$), self-duality ($\mathcal{NC}_3$), training accuracy, and test accuracy. The results in Figure 10 are consistent with those in Figure 5, implying that the fixed classifier and setting $d = K$ exhibits the same within-class variation collapse for the feature $\boldsymbol{H}$, and achieves the same classification accuracy as the *fully-trained* classifier with $d > K$.

## C    Basics

**Definition C.1 ($K$-Simplex ETF)** *A standard Simplex ETF is a collection of points in $\mathbb{R}^K$ specified by the columns of*

$$\boldsymbol{M} = \sqrt{\frac{K}{K-1}} \left( \boldsymbol{I}_K - \frac{1}{K} \mathbf{1}_K \mathbf{1}_K^\top \right),$$

where $\boldsymbol{I}_K \in \mathbb{R}^{K \times K}$ is the identity matrix, and $\boldsymbol{1}_K \in \mathbb{R}^K$ is the all ones vector. In the other words, we also have

$$\boldsymbol{M}^\top \boldsymbol{M} \;=\; \boldsymbol{M}\boldsymbol{M}^\top \;=\; \frac{K}{K-1}\left(\boldsymbol{I}_K - \frac{1}{K}\boldsymbol{1}_K \boldsymbol{1}_K^\top\right).$$

As in [1, 26], in this paper we consider general Simplex ETF as a collection of points in $\mathbb{R}^d$ specified by the columns of $\sqrt{\frac{K}{K-1}}\boldsymbol{P}\left(\boldsymbol{I}_K - \frac{1}{K}\boldsymbol{1}_K \boldsymbol{1}_K^\top\right)$, where $(i)$ when $d \geq K$, $\boldsymbol{P} \in \mathbb{R}^{d \times K}$ is an orthonormal matrix, i.e., $\boldsymbol{P}^\top \boldsymbol{P} = \boldsymbol{I}_K$, and $(ii)$ when $d = K-1$, $\boldsymbol{P}$ is chosen such that $\begin{bmatrix} \boldsymbol{P}^\top & \frac{1}{\sqrt{K}}\boldsymbol{1}_K \end{bmatrix}$ is an orthonormal matrix.

**Lemma C.2 (Young's Inequality)** Let $p, q$ be positive real numbers satisfying $\frac{1}{p} + \frac{1}{q} = 1$. Then for any $a, b \in \mathbb{R}$, we have

$$|ab| \;\leq\; \frac{|a|^p}{p} \;+\; \frac{|b|^q}{q},$$

where the equality holds if and only if $|a|^p = |b|^q$. The case $p = q = 2$ is just the AM-GM inequality for $a^2$, $b^2$: $|ab| \leq \frac{1}{2}\left(a^2 + b^2\right)$, where the equality holds if and only if $|a| = |b|$.

The following Lemma extends the standard variational form of the nuclear norm.

**Lemma C.3** For any fixed $\boldsymbol{Z} \in \mathbb{R}^{K \times N}$ and $\alpha > 0$, we have

$$\|\boldsymbol{Z}\|_* \;=\; \min_{\boldsymbol{Z} = \boldsymbol{W}\boldsymbol{H}} \frac{1}{2\sqrt{\alpha}}\left(\|\boldsymbol{W}\|_F^2 + \alpha\|\boldsymbol{H}\|_F^2\right). \tag{10}$$

Here, $\|\boldsymbol{Z}\|_*$ denotes the nuclear norm of $\boldsymbol{Z}$:

$$\|\boldsymbol{Z}\|_* \;:=\; \sum_{k=1}^{\min\{K,N\}} \sigma_k(\boldsymbol{Z}) = \operatorname{trace}\left(\boldsymbol{\Sigma}\right), \quad \text{with} \quad \boldsymbol{Z} \;=\; \boldsymbol{U}\boldsymbol{\Sigma}\boldsymbol{V}^\top,$$

where $\{\sigma_k\}_{k=1}^{\min\{K,N\}}$ denotes the singular values of $\boldsymbol{Z}$, and $\boldsymbol{Z} = \boldsymbol{U}\boldsymbol{\Sigma}\boldsymbol{V}^\top$ is the singular value decomposition (SVD) of $\boldsymbol{Z}$.

**Proof** [Proof of Lemma C.3] Let $\boldsymbol{Z} = \boldsymbol{U}\boldsymbol{\Sigma}\boldsymbol{V}^\top$ be the SVD of $\boldsymbol{Z}$. First of all, by the facts that $\boldsymbol{U}^\top \boldsymbol{U} = \boldsymbol{I}$, $\boldsymbol{V}^\top \boldsymbol{V} = \boldsymbol{I}$, $\operatorname{trace}\boldsymbol{A}^\top \boldsymbol{A} = \|\boldsymbol{A}\|_F^2$ for any $\boldsymbol{A} \in \mathbb{R}^{n_1 \times n_2}$, and cyclic permutation invariance of $\operatorname{trace}\left(\cdot\right)$, we have

$$\|\boldsymbol{Z}\|_* \;=\; \operatorname{trace}\left(\boldsymbol{\Sigma}\right) \;=\; \frac{1}{2\sqrt{\alpha}}\operatorname{trace}\left(\sqrt{\alpha}\boldsymbol{U}^\top \boldsymbol{U}\boldsymbol{\Sigma}\right) + \frac{\sqrt{\alpha}}{2}\operatorname{trace}\left(\frac{1}{\sqrt{\alpha}}\boldsymbol{\Sigma}\boldsymbol{V}^\top \boldsymbol{V}\right)$$

$$=\; \frac{1}{2\sqrt{\alpha}}\left(\left\|\alpha^{1/4}\boldsymbol{U}\boldsymbol{\Sigma}^{1/2}\right\|_F^2 + \alpha\left\|\alpha^{-1/4}\boldsymbol{\Sigma}^{1/2}\boldsymbol{V}^\top\right\|_F^2\right).$$

This implies that there exists some $\boldsymbol{W} = \alpha^{1/4}\boldsymbol{U}\boldsymbol{\Sigma}^{1/2}$ and $\boldsymbol{H} = \alpha^{-1/4}\boldsymbol{\Sigma}^{1/2}\boldsymbol{V}^\top$, such that $\|\boldsymbol{Z}\|_* = \frac{1}{2\sqrt{\alpha}}\left(\|\boldsymbol{W}\|_F^2 + \alpha\|\boldsymbol{H}\|_F^2\right)$. This equality further implies that

$$\|\boldsymbol{Z}\|_* \;\geq\; \min_{\boldsymbol{Z} = \boldsymbol{W}\boldsymbol{H}} \frac{1}{2\sqrt{\alpha}}\left(\|\boldsymbol{W}\|_F^2 + \alpha\|\boldsymbol{H}\|_F^2\right). \tag{11}$$

On the other hand, for any $\boldsymbol{W}\boldsymbol{H} = \boldsymbol{Z}$, we have

$$\|\boldsymbol{Z}\|_* \;=\; \operatorname{trace}\left(\boldsymbol{\Sigma}\right) \;=\; \operatorname{trace}\left(\boldsymbol{U}^\top \boldsymbol{Z}\boldsymbol{V}\right) \;=\; \operatorname{trace}\left(\boldsymbol{U}^\top \boldsymbol{W}\boldsymbol{H}\boldsymbol{V}\right)$$

$$\leq\; \frac{1}{2\sqrt{\alpha}}\left\|\boldsymbol{U}^\top \boldsymbol{W}\right\|_F^2 + \frac{\sqrt{\alpha}}{2}\left\|\boldsymbol{H}\boldsymbol{V}\right\|_F^2 \;\leq\; \frac{1}{2\sqrt{\alpha}}\left(\|\boldsymbol{W}\|_F^2 + \alpha\|\boldsymbol{H}\|_F^2\right),$$

where the first inequality utilize the Young's inequality in Lemma C.2 that $|\operatorname{trace}(\boldsymbol{A}\boldsymbol{B})| \leq \frac{1}{2c}\|\boldsymbol{A}\|_F^2 + \frac{c}{2}\|\boldsymbol{B}\|_F^2$ for any $c > 0$ and $\boldsymbol{A}, \boldsymbol{B}$ of appropriate dimensions, and the last inequality follows because $\|\boldsymbol{U}\| = 1$ and $\|\boldsymbol{V}\| = 1$. Therefore, we have

$$\|\boldsymbol{Z}\|_* \;\leq\; \min_{\boldsymbol{Z} = \boldsymbol{W}\boldsymbol{H}} \frac{1}{2\sqrt{\alpha}}\left(\|\boldsymbol{W}\|_F^2 + \alpha\|\boldsymbol{H}\|_F^2\right). \tag{12}$$

Combining the results in (11) and (12), we complete the proof. ∎

**Lemma C.4 (Theorem 7.2.6 of [114])** *Let $\boldsymbol{A} \in \mathbb{R}^{n \times n}$ be a symmetric positive semidefinite matrix. Then for any fixed $k \in \{2, 3, \cdots\}$, there exists a unique real symmetric positive semidefinte matrix $\boldsymbol{B}$ such that $\boldsymbol{B}^k = \boldsymbol{A}$.*

# D  Proof of Theorem 3.1

In this part of appendices, we prove Theorem 3.1 in Section 3 that we restate as follows.

**Theorem D.1 (Global Optimality Condition)** *Assume that the feature dimension $d$ is larger than the number of classes $K$, i.e., $d \geq K - 1$. Then any global minimizer $(\boldsymbol{W}^\star, \boldsymbol{H}^\star, \boldsymbol{b}^\star)$ of*

$$\min_{\boldsymbol{W}, \boldsymbol{H}, \boldsymbol{b}} \ f(\boldsymbol{W}, \boldsymbol{H}, \boldsymbol{b}) \ := \ g(\boldsymbol{W}\boldsymbol{H} + \boldsymbol{b}\mathbf{1}^\top) \ + \ \frac{\lambda_{\boldsymbol{W}}}{2} \|\boldsymbol{W}\|_F^2 + \frac{\lambda_{\boldsymbol{H}}}{2} \|\boldsymbol{H}\|_F^2 + \frac{\lambda_{\boldsymbol{b}}}{2} \|\boldsymbol{b}\|_2^2 \quad (13)$$

*with*

$$g(\boldsymbol{W}\boldsymbol{H} + \boldsymbol{b}\mathbf{1}^\top) := \frac{1}{N} \sum_{k=1}^{K} \sum_{i=1}^{n} \mathcal{L}_{\mathrm{CE}}(\boldsymbol{W}\boldsymbol{h}_{k,i} + \boldsymbol{b}, \boldsymbol{y}_k), \quad (14)$$

*obeys the following*

$$\|\boldsymbol{w}^\star\|_2 \ = \ \left\|\boldsymbol{w}^{\star 1}\right\|_2 \ = \ \left\|\boldsymbol{w}^{\star 2}\right\|_2 \ = \ \cdots \ = \ \left\|\boldsymbol{w}^{\star K}\right\|_2, \quad \text{and} \quad \boldsymbol{b}^\star = b^\star \mathbf{1},$$

$$\boldsymbol{h}_{k,i}^\star \ = \ \sqrt{\frac{\lambda_{\boldsymbol{W}}}{\lambda_{\boldsymbol{H}} n}} \boldsymbol{w}^{\star k}, \quad \forall \, k \in [K], \ i \in [n], \quad \text{and} \quad \overline{\boldsymbol{h}}_i^\star := \frac{1}{K} \sum_{j=1}^{K} \boldsymbol{h}_{j,i}^\star \ = \ \mathbf{0}, \quad \forall \, i \in [n],$$

*where either $b^\star = 0$ or $\lambda_{\boldsymbol{b}} = 0$, and the matrix $\frac{1}{\|\boldsymbol{w}^\star\|_2} \boldsymbol{W}^{\star\top}$ forms a $K$-simplex ETF defined in Definition C.1 in the sense that*

$$\frac{1}{\|\boldsymbol{w}^\star\|_2^2} \boldsymbol{W}^{\star\top} \boldsymbol{W}^\star \ = \ \frac{K}{K-1} \left( \boldsymbol{I}_K - \frac{1}{K} \mathbf{1}_K \mathbf{1}_K^\top \right).$$

## D.1  Main Proof

Similar to the proofs in [24, 26], we prove the theorem by directly showing that $f(\boldsymbol{W}, \boldsymbol{H}, \boldsymbol{b}) > f(\boldsymbol{W}^\star, \boldsymbol{H}^\star, \boldsymbol{b}^\star)$ for any $(\boldsymbol{W}, \boldsymbol{H}, \boldsymbol{b})$ not in the form as shown in Theorem D.1.

**Proof** [Proof of Theorem D.1] First note that the objective function $f$ is *coercive*[19] due to the weight decay regularizers and the fact that the CE loss is always non-negative. This implies that the global minimizer of $f(\boldsymbol{W}, \boldsymbol{H}, \boldsymbol{b})$ in (13) is always finite. By Lemma D.2, we know that any critical point $(\boldsymbol{W}, \boldsymbol{H}, \boldsymbol{b})$ of $f$ in (13) satisfies

$$\boldsymbol{W}^\top \boldsymbol{W} = \frac{\lambda_{\boldsymbol{H}}}{\lambda_{\boldsymbol{W}}} \boldsymbol{H}\boldsymbol{H}^\top.$$

For the rest of the proof, to simplify the notations, let $\|\boldsymbol{W}\|_F^2 = \rho$, and thus $\|\boldsymbol{H}\|_F^2 = \frac{\lambda_{\boldsymbol{H}}}{\lambda_{\boldsymbol{W}}}\rho$.

We will first provide a lower bound for the cross-entropy term $g(\boldsymbol{W}\boldsymbol{H} + \boldsymbol{b}\mathbf{1}^\top)$ for any $\boldsymbol{W}$ with energy $\rho$, and then show that the lower bound is attained if and only if the parameters are in the form described in Theorem D.1. Now, by Lemma D.3, we know that for any $c_1, c_3 > 0$,

$$g(\boldsymbol{W}\boldsymbol{H} + \boldsymbol{b}\mathbf{1}^\top) \ \geq \ -\frac{\rho}{(1 + c_1)(K-1)} \sqrt{\frac{\lambda_{\boldsymbol{W}}}{\lambda_{\boldsymbol{H}} n}} \ + \ c_2$$

with $c_2 = \frac{1}{1+c_1} \log\left((1 + c_1)(K-1)\right) + \frac{c_1}{1+c_1} \log\left(\frac{1+c_1}{c_1}\right)$. Therefore, we have

$$f(\boldsymbol{W}, \boldsymbol{H}, \boldsymbol{b}) \ = \ g(\boldsymbol{W}\boldsymbol{H} + \boldsymbol{b}\mathbf{1}^\top) \ + \ \frac{\lambda_{\boldsymbol{W}}}{2} \|\boldsymbol{W}\|_F^2 + \frac{\lambda_{\boldsymbol{H}}}{2} \|\boldsymbol{H}\|_F^2 + \frac{\lambda_{\boldsymbol{b}}}{2} \|\boldsymbol{b}\|_2^2$$

$$\geq \ \underbrace{-\frac{\rho}{(1 + c_1)(K-1)} \sqrt{\frac{\lambda_{\boldsymbol{W}}}{\lambda_{\boldsymbol{H}} n}} + c_2 + \lambda_{\boldsymbol{W}}\rho}_{\xi(\rho, \lambda_{\boldsymbol{W}}, \lambda_{\boldsymbol{H}})} + \frac{\lambda_{\boldsymbol{b}}}{2} \|\boldsymbol{b}\|_2^2$$

$$\geq \ \xi\left(\rho, \lambda_{\boldsymbol{W}}, \lambda_{\boldsymbol{H}}\right),$$

---

[19] A function $f : \mathbb{R}^n \mapsto \mathbb{R}$ is coercive if $f(\boldsymbol{x}) \to +\infty$ as $\|\boldsymbol{x}\|_2 \to +\infty$.

where the last inequality becomes an equality whenever either $\lambda_{\boldsymbol{b}} = 0$ or $\boldsymbol{b} = \boldsymbol{0}$. Furthermore, by Lemma D.4, we know that the inequality $f(\boldsymbol{W}, \boldsymbol{H}, \boldsymbol{b}) \geq \xi(\rho, \lambda_{\boldsymbol{W}}, \lambda_{\boldsymbol{H}})$ becomes an equality *if and only if* $(\boldsymbol{W}, \boldsymbol{H}, \boldsymbol{b})$ satisfy the following

(a) $\|\boldsymbol{w}\|_2 = \|\boldsymbol{w}^1\|_2 = \|\boldsymbol{w}^2\|_2 = \cdots = \|\boldsymbol{w}^K\|_2$;

(b) $\boldsymbol{b} = b\mathbf{1}$, where either $b = 0$ or $\lambda_{\boldsymbol{b}} = 0$;

(c) $\overline{\boldsymbol{h}}_i := \frac{1}{K}\sum_{j=1}^K \boldsymbol{h}_{j,i} = \boldsymbol{0}, \quad \forall\, i \in [n]$, and $\sqrt{\frac{\lambda_{\boldsymbol{W}}}{\lambda_{\boldsymbol{H}} n}}\boldsymbol{w}^k = \boldsymbol{h}_{k,i}, \quad \forall\, k \in [K],\ i \in [n]$;

(d) $\boldsymbol{W}\boldsymbol{W}^\top = \frac{\rho}{K-1}\left(\boldsymbol{I}_K - \frac{1}{K}\mathbf{1}_K\mathbf{1}_K^\top\right)$;

(e) $c_1 = \left[(K-1)\exp\left(-\frac{\rho}{K-1}\sqrt{\frac{\lambda_{\boldsymbol{W}}}{\lambda_{\boldsymbol{H}} n}}\right)\right]^{-1}$.

To finish the proof, we only need to show that $\rho = \|\boldsymbol{W}\|_F^2$ must be finite for any fixed $\lambda_{\boldsymbol{W}}, \lambda_{\boldsymbol{H}} > 0$. From (e), we know that $c_1 = \left[(K-1)\exp\left(-\frac{\rho}{K-1}\sqrt{\frac{\lambda_{\boldsymbol{W}}}{\lambda_{\boldsymbol{H}} n}}\right)\right]^{-1}$ is an *increasing* function in terms of $\rho$, and $c_2 = \frac{1}{1+c_1}\log\left((1+c_1)(K-1)\right) + \frac{c_1}{1+c_1}\log\left(\frac{1+c_1}{c_1}\right)$ is a *decreasing* function in terms of $c_1$. Therefore, we observe the following:

- When $\rho \to 0$, we have $c_1 \to \frac{1}{K-1}$ and $c_2 \to \log K$, so that

$$\lim_{\rho \to 0} \xi(\rho; \lambda_{\boldsymbol{W}}, \lambda_{\boldsymbol{H}}) = \lim_{\rho \to 0} c_2(\rho) = \log K.$$

- On the other hand, when $\rho \to +\infty$, $c_1 \to +\infty$ and $c_2 \to 0$, so that $\xi(\rho; \lambda_{\boldsymbol{W}}, \lambda_{\boldsymbol{H}}) \to +\infty$ as $\rho \to +\infty$.

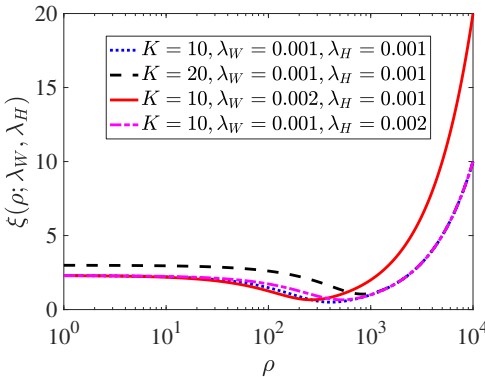

Figure 11: Plot of $\xi(\rho; \lambda_{\boldsymbol{W}}, \lambda_{\boldsymbol{H}})$ in terms of $\rho$ for $n = 100$ and different $K, \lambda_{\boldsymbol{W}}, \lambda_{\boldsymbol{H}}$.

Since $\xi(\rho; \lambda_{\boldsymbol{W}}, \lambda_{\boldsymbol{H}})$ is a continuous function of $\rho \in [0, +\infty)$ and $\xi(\rho; \lambda_{\boldsymbol{W}}, \lambda_{\boldsymbol{H}}) \to +\infty$ and $\rho \to +\infty$, these further imply that $\xi(\rho; \lambda_{\boldsymbol{W}}, \lambda_{\boldsymbol{H}})$ achieves its minimum at a finite $\rho$ (see Figure 11 for an example). This finishes the proof. ∎

### D.2 Supporting Lemmas

We first characterize the following balance property between $\boldsymbol{W}$ and $\boldsymbol{H}$ for any critical point $(\boldsymbol{W}, \boldsymbol{H}, \boldsymbol{b})$ of our loss function:

**Lemma D.2** *Let $\rho = \|\boldsymbol{W}\|_F^2$. Any critical point $(\boldsymbol{W}, \boldsymbol{H}, \boldsymbol{b})$ of (13) obeys*

$$\boldsymbol{W}^\top\boldsymbol{W} = \frac{\lambda_{\boldsymbol{H}}}{\lambda_{\boldsymbol{W}}}\boldsymbol{H}\boldsymbol{H}^\top \quad and \quad \rho = \|\boldsymbol{W}\|_F^2 = \frac{\lambda_{\boldsymbol{H}}}{\lambda_{\boldsymbol{W}}}\|\boldsymbol{H}\|_F^2. \tag{15}$$

**Proof** [Proof of Lemma D.2] By definition, any critical point $(\boldsymbol{W}, \boldsymbol{H}, \boldsymbol{b})$ of (13) satisfies the following:

$$\nabla_{\boldsymbol{W}} f(\boldsymbol{W}, \boldsymbol{H}, \boldsymbol{b}) = \nabla_{\boldsymbol{Z}=\boldsymbol{W}\boldsymbol{H}} \, g(\boldsymbol{W}\boldsymbol{H} + \boldsymbol{b}\boldsymbol{1}^\top) \boldsymbol{H}^\top + \lambda_{\boldsymbol{W}} \boldsymbol{W} = \boldsymbol{0}, \qquad (16)$$

$$\nabla_{\boldsymbol{H}} f(\boldsymbol{W}, \boldsymbol{H}, \boldsymbol{b}) = \boldsymbol{W}^\top \nabla_{\boldsymbol{Z}=\boldsymbol{W}\boldsymbol{H}} \, g(\boldsymbol{W}\boldsymbol{H} + \boldsymbol{b}\boldsymbol{1}^\top) + \lambda_{\boldsymbol{H}} \boldsymbol{H} = \boldsymbol{0}. \qquad (17)$$

Left multiply the first equation by $\boldsymbol{W}^\top$ on both sides and then right multiply second equation by $\boldsymbol{H}^\top$ on both sides, it gives

$$\boldsymbol{W}^\top \nabla_{\boldsymbol{Z}=\boldsymbol{W}\boldsymbol{H}} \, g(\boldsymbol{W}\boldsymbol{H} + \boldsymbol{b}\boldsymbol{1}^\top) \boldsymbol{H}^\top = -\lambda_{\boldsymbol{W}} \boldsymbol{W}^\top \boldsymbol{W},$$

$$\boldsymbol{W}^\top \nabla_{\boldsymbol{Z}=\boldsymbol{W}\boldsymbol{H}} \, g(\boldsymbol{W}\boldsymbol{H} + \boldsymbol{b}\boldsymbol{1}^\top) \boldsymbol{H}^\top = -\lambda_{\boldsymbol{H}} \boldsymbol{H}^\top \boldsymbol{H}.$$

Therefore, combining the equations above, we obtain

$$\lambda_{\boldsymbol{W}} \boldsymbol{W}^\top \boldsymbol{W} = \lambda_{\boldsymbol{H}} \boldsymbol{H}\boldsymbol{H}^\top.$$

Moreover, we have

$$\rho = \|\boldsymbol{W}\|_F^2 = \mathrm{trace}\left(\boldsymbol{W}^\top \boldsymbol{W}\right) = \frac{\lambda_{\boldsymbol{H}}}{\lambda_{\boldsymbol{W}}} \mathrm{trace}\left(\boldsymbol{H}\boldsymbol{H}^\top\right) = \frac{\lambda_{\boldsymbol{H}}}{\lambda_{\boldsymbol{W}}} \mathrm{trace}\left(\boldsymbol{H}^\top \boldsymbol{H}\right) = \frac{\lambda_{\boldsymbol{H}}}{\lambda_{\boldsymbol{W}}} \|\boldsymbol{H}\|_F^2,$$

as desired. ∎

**Lemma D.3** *Let* $\boldsymbol{W} = \begin{bmatrix} (\boldsymbol{w}^1)^\top \\ \vdots \\ (\boldsymbol{w}^K)^\top \end{bmatrix} \in \mathbb{R}^{K \times d}$, $\boldsymbol{H} = [\boldsymbol{h}_{1,1} \cdots \boldsymbol{h}_{K,n}] \in \mathbb{R}^{d \times N}$, $N = nK$, *and*
$\rho = \|\boldsymbol{W}\|_F^2$. *Given* $g(\boldsymbol{W}\boldsymbol{H} + \boldsymbol{b}\boldsymbol{1}^\top)$ *defined in* (14), *for any critical point* $(\boldsymbol{W}, \boldsymbol{H}, \boldsymbol{b})$ *of* (13) *and* $c_1 > 0$, *we have*

$$g(\boldsymbol{W}\boldsymbol{H} + \boldsymbol{b}\boldsymbol{1}^\top) \geq -\frac{\rho}{(1 + c_1)(K - 1)} \sqrt{\frac{\lambda_{\boldsymbol{W}}}{\lambda_{\boldsymbol{H}} n}} + c_2, \qquad (18)$$

*with* $c_2 = \frac{1}{1+c_1} \log\left((1 + c_1)(K - 1)\right) + \frac{c_1}{1+c_1} \log\left(\frac{1+c_1}{c_1}\right)$.

**Proof** [Proof of Lemma D.3] By Lemma D.5 with $\boldsymbol{z}_{k,i} = \boldsymbol{W}\boldsymbol{h}_{k,i} + \boldsymbol{b}$, since the scalar $c_1 > 0$ can be arbitrary, we choose the same $c_1$ and $c_2$ for all $i \in [n]$ and $k \in [K]$, we have the following lower bound for $g(\boldsymbol{W}\boldsymbol{H} + \boldsymbol{b}\boldsymbol{1}^\top)$ as

$$(1 + c_1)(K - 1) \left[ g(\boldsymbol{W}\boldsymbol{H} + \boldsymbol{b}\boldsymbol{1}^\top) - c_2 \right]$$

$$= (1 + c_1)(K - 1) \left[ \frac{1}{N} \sum_{k=1}^{K} \sum_{i=1}^{n} \mathcal{L}_{\mathrm{CE}}(\boldsymbol{W}\boldsymbol{h}_{k,i} + \boldsymbol{b}, \boldsymbol{y}_k) - c_2 \right]$$

$$\geq \frac{1}{N} \sum_{k=1}^{K} \sum_{i=1}^{n} \left[ \sum_{j=1}^{K} \left(\boldsymbol{h}_{k,i}^\top \boldsymbol{w}^j + b_j\right) - K\left(\boldsymbol{h}_{k,i}^\top \boldsymbol{w}^k + b_k\right) \right] \qquad (19)$$

$$= \frac{1}{N} \sum_{i=1}^{n} \left[ \left( \sum_{k=1}^{K} \sum_{j=1}^{K} \boldsymbol{h}_{k,i}^\top \boldsymbol{w}^j - K \sum_{k=1}^{K} \boldsymbol{h}_{k,i}^\top \boldsymbol{w}^k \right) + \underbrace{\sum_{k=1}^{K} \sum_{j=1}^{K} (b_j - b_k)}_{=0} \right]$$

$$= \frac{1}{N} \sum_{i=1}^{n} \left( \sum_{k=1}^{K} \sum_{j=1}^{K} \boldsymbol{h}_{j,i}^\top \boldsymbol{w}^k - K \sum_{k=1}^{K} \boldsymbol{h}_{k,i}^\top \boldsymbol{w}^k \right)$$

$$= \frac{K}{N} \sum_{i=1}^{n} \sum_{k=1}^{K} \left[ \left( \frac{1}{K} \sum_{j=1}^{K} (\boldsymbol{h}_{j,i} - \boldsymbol{h}_{k,i}) \right)^\top \boldsymbol{w}^k \right] = \frac{1}{n} \sum_{i=1}^{n} \sum_{k=1}^{K} \left( \overline{\boldsymbol{h}}_i - \boldsymbol{h}_{k,i} \right)^\top \boldsymbol{w}^k,$$

where for the last equality we let $\overline{h}_i = \frac{1}{K}\sum_{j=1}^{K} h_{j,i}$. Furthermore, from the AM-GM inequality in Lemma C.2, we know that for any $u, v \in \mathbb{R}^K$ and any $c_3 > 0$,

$$u^\top v \;\leq\; \frac{c_3}{2}\,\|u\|_2^2 \;+\; \frac{1}{2c_3}\,\|v\|_2^2\,,$$

where the inequality becomes an equality when $c_3 u = v$. Thus, we further have

$$(1+c_1)(K-1)\left[g(WH + b\mathbf{1}^\top) - c_2\right]$$

$$\geq -\frac{c_3}{2}\sum_{k=1}^{K}\left\|w^k\right\|_2^2 \;-\; \frac{1}{2c_3 n}\sum_{i=1}^{n}\sum_{k=1}^{K}\left\|\overline{h}_i - h_{k,i}\right\|_2^2$$

$$= -\frac{c_3}{2}\sum_{k=1}^{K}\left\|w^k\right\|_2^2 \;-\; \frac{1}{2c_3 n}\sum_{i=1}^{n}\left[\left(\sum_{k=1}^{K}\left\|h_{k,i}\right\|_2^2\right) - K\left\|\overline{h}_i\right\|_2^2\right]$$

$$= -\frac{c_3}{2}\left\|W\right\|_F^2 \;-\; \frac{1}{2c_3 n}\left(\left\|H\right\|_F^2 - K\sum_{i=1}^{n}\left\|\overline{h}_i\right\|_2^2\right),$$

where the first inequality becomes an equality if and only if

$$c_3 w^k = (h_{k,i} - \overline{h}_i), \quad \forall\, k \in [K],\ i \in [n]. \tag{20}$$

Let $\rho = \|W\|_F^2$. Now, by using Lemma D.2, we have $W^\top W = \frac{\lambda_H}{\lambda_W} HH^\top$, so that $\|H\|_F^2 = \mathrm{trace}\left(HH^\top\right) = \frac{\lambda_W}{\lambda_H}\,\mathrm{trace}\left(W^\top W\right) = \frac{\lambda_W}{\lambda_H}\rho$. Therefore, we have

$$g(WH + b\mathbf{1}^\top) \geq -\frac{\rho}{2(1+c_1)(K-1)}\left(c_3 + \frac{\lambda_W}{\lambda_H}\frac{1}{c_3 n}\right) + c_2, \tag{21}$$

as desired. The last inequality achieves its equality if and only if

$$\overline{h}_i = \mathbf{0}, \quad \forall\, i \in [n]. \tag{22}$$

Plugging this into (20), we have

$$c_3 w^k \;=\; h_{k,i} \Rightarrow c_3^2 \;=\; \frac{\sum_{i=1}^{n}\sum_{k=1}^{K}\|h_{k,i}\|_2^2}{n\sum_{k=1}^{K}\|w^k\|_2^2} \;=\; \frac{\|H\|_F^2}{n\,\|W\|_F^2} \;=\; \frac{\lambda_W}{n\lambda_H}.$$

This together with the lower bound in (21) gives

$$g(WH + b\mathbf{1}^\top) \geq -\frac{\rho}{(1+c_1)(K-1)}\sqrt{\frac{\lambda_W}{\lambda_H n}} + c_2,$$

as suggested in (18). $\blacksquare$

Next, we show that the lower bound in (18) is attained if and only if $(W, H, b)$ satisfies the following conditions.

**Lemma D.4** *Under the same assumptions of Lemma D.3, the lower bound in (18) is attained for any critical point $(W, H, b)$ of (13) if and only if the following hold*

$$\left\|w^1\right\|_2 \;=\; \left\|w^2\right\|_2 \;=\; \cdots \;=\; \left\|w^K\right\|_2, \quad and \quad b = b\mathbf{1},$$

$$\overline{h}_i \;:=\; \frac{1}{K}\sum_{j=1}^{K} h_{j,i} \;=\; \mathbf{0}, \quad \forall\, i \in [n], \quad and \quad \sqrt{\frac{\lambda_W}{\lambda_H n}}\,w^k \;=\; h_{k,i}, \quad \forall\, k \in [K],\ i \in [n],$$

$$WW^\top \;=\; \frac{\rho}{K-1}\left(I_K - \frac{1}{K}\mathbf{1}_K\mathbf{1}_K^\top\right), \quad and\ c_1 \;=\; \left[(K-1)\exp\left(-\frac{\rho}{K-1}\sqrt{\frac{\lambda_W}{\lambda_H n}}\right)\right]^{-1}.$$

The proof of Lemma D.4 utilizes the conditions in Lemma D.5, and the conditions (20) and (22) during the proof of Lemma D.3.

**Proof** [Proof of Lemma D.4] From the proof of D.3, if we want to attain the lower bound, we know that we need at least (20) and (22) to hold, which is equivalent to the following:

$$\overline{\boldsymbol{h}}_i \;=\; \frac{1}{K}\sum_{j=1}^{K}\boldsymbol{h}_{j,i} \;=\; \boldsymbol{0}, \quad \forall i \in [n], \quad \text{and} \quad \sqrt{\frac{\lambda_{\boldsymbol{W}}}{\lambda_{\boldsymbol{H}}n}}\boldsymbol{w}^k \;=\; \boldsymbol{h}_{k,i}, \quad \forall\, k \in [K],\ i \in [n], \quad (23)$$

which further implies that

$$\sum_{k=1}^{K}\boldsymbol{w}^k \;=\; \boldsymbol{0}. \tag{24}$$

Next, under the condition (23), if we want (18) to become an equality, we only need (19) to become an equality, which is true if and only if the condition (34) in Lemma D.5 holds for $\boldsymbol{z}_{k,i}=\boldsymbol{W}\boldsymbol{h}_{k,i}+\boldsymbol{b}$ for all $i\in[n]$ and $k\in[K]$. First, let $[\boldsymbol{z}_{k,i}]_j=\boldsymbol{h}_{k,i}^\top\boldsymbol{w}^j+b_j$, where we have

$$\sum_{j=1}^{K}[\boldsymbol{z}_{k,i}]_j \;=\; \boldsymbol{h}_{k,i}^\top\sum_{j=1}^{K}\boldsymbol{w}^j + \sum_{j=1}^{K}b_j \;=\; \sqrt{\frac{\lambda_{\boldsymbol{H}}n}{\lambda_{\boldsymbol{W}}}}\boldsymbol{h}_{k,i}^\top\sum_{j=1}^{K}\boldsymbol{h}_{j,i} + \sum_{j=1}^{K}b_j$$

$$= \sqrt{\frac{\lambda_{\boldsymbol{H}}n}{\lambda_{\boldsymbol{W}}}}K\boldsymbol{h}_{k,i}^\top\overline{\boldsymbol{h}}_i + \sum_{j=1}^{K}b_j \;=\; K\overline{b} \tag{25}$$

with $\overline{b}=\frac{1}{K}\sum_{i=1}^{K}b_i$, and

$$K[\boldsymbol{z}_{k,i}]_k \;=\; K\boldsymbol{h}_{k,i}^\top\boldsymbol{w}^k + Kb_k \;=\; \sqrt{\frac{\lambda_{\boldsymbol{W}}}{\lambda_{\boldsymbol{H}}n}}\left(K\left\|\boldsymbol{w}^k\right\|_2^2\right) + Kb_k. \tag{26}$$

Based on (25), (26), and (34) from Lemma D.5, we have

$$c_1 \;=\; \left[(K-1)\exp\left(\frac{\left(\sum_{j=1}^{K}[\boldsymbol{z}_{k,i}]_j\right) - K[\boldsymbol{z}_{k,i}]_k}{K-1}\right)\right]^{-1}$$

$$= \left[(K-1)\exp\left(\frac{K}{K-1}\left(\overline{b} - \sqrt{\frac{\lambda_{\boldsymbol{W}}}{\lambda_{\boldsymbol{H}}n}}\left\|\boldsymbol{w}^k\right\|_2^2 - b_k\right)\right)\right]^{-1}. \tag{27}$$

Since the scalar $c_1 > 0$ is chosen to be the same for all $k \in [K]$, we have

$$\sqrt{\frac{\lambda_{\boldsymbol{W}}}{\lambda_{\boldsymbol{H}}n}}\left\|\boldsymbol{w}^k\right\|_2^2 + b_k \;=\; \sqrt{\frac{\lambda_{\boldsymbol{W}}}{\lambda_{\boldsymbol{H}}n}}\left\|\boldsymbol{w}^\ell\right\|_2^2 + b_\ell, \quad \forall\,\ell \neq k. \tag{28}$$

Second, since $[\boldsymbol{z}_{k,i}]_j=[\boldsymbol{z}_{k,i}]_\ell$ for all $\forall j,\ell\neq k,\ k\in[K]$, from (23) we have

$$\boldsymbol{h}_{k,i}^\top\boldsymbol{w}^j + b_j \;=\; \boldsymbol{h}_{k,i}^\top\boldsymbol{w}^\ell + b_\ell, \quad \forall j,\ell\neq k,\ k\in[K]$$

$$\iff \sqrt{\frac{\lambda_{\boldsymbol{W}}}{\lambda_{\boldsymbol{H}}n}}(\boldsymbol{w}^k)^\top\boldsymbol{w}^j + b_j \;=\; \sqrt{\frac{\lambda_{\boldsymbol{W}}}{\lambda_{\boldsymbol{H}}n}}(\boldsymbol{w}^k)^\top\boldsymbol{w}^\ell + b_\ell, \quad \forall j,\ell\neq k,\ k\in[K]. \tag{29}$$

Based on this and (24), we have

$$\sqrt{\frac{\lambda_{\boldsymbol{W}}}{\lambda_{\boldsymbol{H}}n}}\left\|\boldsymbol{w}^k\right\|_2^2 + b_k \;=\; -\sqrt{\frac{\lambda_{\boldsymbol{W}}}{\lambda_{\boldsymbol{H}}n}}\sum_{j\neq k}(\boldsymbol{w}^j)^\top\boldsymbol{w}^k + b_k$$

$$= -(K-1)\sqrt{\frac{\lambda_{\boldsymbol{W}}}{\lambda_{\boldsymbol{H}}n}}\underbrace{(\boldsymbol{w}^\ell)^\top\boldsymbol{w}^k}_{\ell\neq k,\ell\in[K]} + \left(b_k + \sum_{j\neq\ell,k}(b_\ell - b_j)\right)$$

$$= -(K-1)\sqrt{\frac{\lambda_{\boldsymbol{W}}}{\lambda_{\boldsymbol{H}}n}}(\boldsymbol{w}^\ell)^\top\boldsymbol{w}^k + \left[2b_k + (K-1)b_\ell - K\overline{b}\right] \tag{30}$$

for all $\ell\neq k$. Combining (28) and (30), for all $k,\ell\in[K]$ with $k\neq\ell$ we have

$$2b_k + (K-1)b_\ell - K\overline{b} \;=\; 2b_\ell + (K-1)b_k - K\overline{b} \quad \iff \quad b_k \;=\; b_\ell,\ \forall\,k\neq\ell.$$

Therefore, we can write $\boldsymbol{b} = b\mathbf{1}_K$ for some $b > 0$. Moreover, since $b_k = b_\ell$ for all $k \neq \ell$, (28) and (29), and (30) further imply that

$$\left\| \boldsymbol{w}^1 \right\|_2 = \left\| \boldsymbol{w}^2 \right\|_2 = \cdots = \left\| \boldsymbol{w}^K \right\|_2, \quad \text{and} \quad \left\| \boldsymbol{w}^k \right\|_2^2 = \frac{1}{K} \left\| \boldsymbol{W} \right\|_F^2 = \frac{\rho}{K}, \tag{31}$$

$$(\boldsymbol{w}^j)^\top \boldsymbol{w}^k = (\boldsymbol{w}^\ell)^\top \boldsymbol{w}^k = -\frac{1}{K-1} \left\| \boldsymbol{w}^k \right\|_2^2 = -\frac{\rho}{K(K-1)}, \quad \forall j, \ell \neq k, \ k \in [K], \tag{32}$$

where (32) is equivalent to

$$\boldsymbol{W}\boldsymbol{W}^\top = \frac{\rho}{K-1} \left( \boldsymbol{I}_K - \frac{1}{K} \mathbf{1}_K \mathbf{1}_K^\top \right).$$

Finally, plugging the results in (31) and (32) into (27), we have

$$c_1 = \left[ (K-1) \exp\left( -\frac{\rho}{K-1} \sqrt{\frac{\lambda_{\boldsymbol{W}}}{\lambda_{\boldsymbol{H}} n}} \right) \right]^{-1},$$

as desired. ■

**Lemma D.5** *Let $\boldsymbol{y}_k \in \mathbb{R}^K$ be an one-hot vector with the $k$th entry equalling $1$ for some $k \in [K]$. For any vector $\boldsymbol{z} \in \mathbb{R}^K$ and $c_1 > 0$, the cross-entropy loss $\mathcal{L}_{\mathrm{CE}}(\boldsymbol{z}, \boldsymbol{y}_k)$ with $\boldsymbol{y}_k$ can be lower bounded by*

$$\mathcal{L}_{\mathrm{CE}}(\boldsymbol{z}, \boldsymbol{y}_k) \geq \frac{1}{1+c_1} \frac{\left( \sum_{i=1}^K z_i \right) - K z_k}{K-1} + c_2, \tag{33}$$

*where $c_2 = \frac{1}{1+c_1} \log\left( (1+c_1)(K-1) \right) + \frac{c_1}{1+c_1} \log\left( \frac{1+c_1}{c_1} \right)$. The inequality becomes an equality when*

$$z_i = z_j, \quad \forall i, j \neq k, \quad \text{and} \quad c_1 = \left[ (K-1) \exp\left( \frac{\left( \sum_{i=1}^K z_i \right) - K z_k}{K-1} \right) \right]^{-1}. \tag{34}$$

**Proof** [Proof of Lemma D.5] Notice that for any vector $\boldsymbol{z} \in \mathbb{R}^K$, the cross-entropy loss with $\boldsymbol{y}_k$ can be lower bounded by

$$\mathcal{L}_{\mathrm{CE}}(\boldsymbol{z}, \boldsymbol{y}_k) = \log\left( \frac{\sum_{i=1}^K \exp(z_i)}{\exp(z_k)} \right) = \log\left( 1 + \sum_{i \neq k} \exp(z_i - z_k) \right)$$

$$\geq \log\left( 1 + (K-1) \exp\left( \sum_{i \neq k} \frac{z_i - z_k}{K-1} \right) \right)$$

$$= \log\left( 1 + (K-1) \exp\left( \frac{\sum_{i=1}^K z_i - K z_k}{K-1} \right) \right) \tag{35}$$

where the inequality follows from the Jensen's inequality that

$$\sum_{i \neq k} \exp(z_i - z_k) = (K-1) \sum_{i \neq k} \frac{1}{K-1} \exp(z_i - z_k) \geq (K-1) \exp\left( \sum_{i \neq k} \frac{z_i - z_k}{K-1} \right),$$

which achieves the equality only when $z_i = z_j$ for all $i, j \neq k$. Second, by the concavity of the $\log(\cdot)$ function (i.e., $\log(tx + (1-t)x') \geq t \log x + (1-t) \log x'$ for any $x, x' \in \mathbb{R}$ and $t \in [0, 1]$,

which becomes an equality iff $x = x'$, or $t = 0$, or $t = 1$), from (35), for any $c_1 > 0$ we have

$$
\mathcal{L}_{\text{CE}}(\boldsymbol{z}, \boldsymbol{y}_k)
$$

$$
= \log \left( \frac{c_1}{1+c_1} \frac{1+c_1}{c_1} + \frac{1}{1+c_1}(1+c_1)(K-1)\exp\left( \frac{\sum_{i=1}^{K} z_i - Kz_k}{K-1} \right) \right)
$$

$$
\geq \frac{1}{1+c_1} \log \left( (1+c_1)(K-1)\exp\left( \frac{\sum_{i=1}^{K} z_i - Kz_k}{K-1} \right) \right) + \frac{c_1}{1+c_1} \log \left( \frac{1+c_1}{c_1} \right)
$$

$$
= \frac{1}{1+c_1} \frac{\left( \sum_{i=1}^{K} z_i \right) - Kz_k}{K-1} + \underbrace{\frac{1}{1+c_1} \log \left( (1+c_1)(K-1) \right) + \frac{c_1}{1+c_1} \log \left( \frac{1+c_1}{c_1} \right)}_{c_2},
$$

as desired. The last inequality becomes an equality if any only if

$$
\frac{1+c_1}{c_1} = (1+c_1)(K-1)\exp\left( \frac{\sum_{i=1}^{K} z_i - Kz_k}{K-1} \right) \quad \text{or} \quad c_1 = 0, \quad \text{or} \quad c_1 = +\infty.
$$

However, when $c_1 = 0$ or $c_1 = +\infty$, the equality is trivial. Therefore, we have

$$
c_1 = \left[ (K-1)\exp\left( \frac{\left( \sum_{i=1}^{K} z_i \right) - Kz_k}{K-1} \right) \right]^{-1},
$$

as desired. ∎

# E Proof of Theorem 3.2

In this part of appendices, we prove Theorem 3.2 in Section 3. In particular, we analyze the global optimization landscape of

$$
\min_{\boldsymbol{W}, \boldsymbol{H}, \boldsymbol{b}} \ f(\boldsymbol{W}, \boldsymbol{H}, \boldsymbol{b}) = \frac{1}{Kn} \sum_{k=1}^{K} \sum_{i=1}^{n} \mathcal{L}_{\text{CE}}(\boldsymbol{W}\boldsymbol{h}_{k,i} + \boldsymbol{b}, \boldsymbol{y}_k) + \frac{\lambda_{\boldsymbol{W}}}{2} \|\boldsymbol{W}\|_F^2 + \frac{\lambda_{\boldsymbol{H}}}{2} \|\boldsymbol{H}\|_F^2 + \frac{\lambda_{\boldsymbol{b}}}{2} \|\boldsymbol{b}\|_2^2,
$$

with respect to $\boldsymbol{W} \in \mathbb{R}^{K \times d}$, $\boldsymbol{H} = [\boldsymbol{h}_{1,1} \cdots \boldsymbol{h}_{K,n}] \in \mathbb{R}^{d \times N}$, and $\boldsymbol{b} \in \mathbb{R}^K$. We show that the function is a strict saddle function [49–51] in the Euclidean space, that there is no spurious local minimizer and all global minima are corresponding to the form showed in Theorem D.1.

**Theorem E.1 (No Spurious Local Minima and Strict Saddle Property)** *Assume that the feature dimension $d$ is larger than the number of classes $K$, i.e., $d > K$. Then the function $f(\boldsymbol{W}, \boldsymbol{H}, \boldsymbol{b})$ in (13) is a strict saddle function with no spurious local minimum:*

- *Any local minimizer of (13) is a global minimum of the form shown in Theorem D.1.*

- *Any critical point of (13) that is not a local minimum has at least one negative curvature direction, i.e., the Hessian $\nabla^2 f(\boldsymbol{W}, \boldsymbol{H}, \boldsymbol{b})$ at this point has at least one negative eigenvalue*

$$
\lambda_i \left( \nabla^2 f(\boldsymbol{W}, \boldsymbol{H}, \boldsymbol{b}) \right) < 0.
$$

## E.1 Main Proof

**Proof** [Proof of Theorem E.1] The main idea of proving Theorem 3.2 is to first connect the problem (13) to its corresponding convex counterpart, so that we can obtain the global optimality conditions for the original problem (13) based on the convex counterpart. Finally, we characterize the properties of all the critical points based on the optimality condition. We describe this in more detail as follows.

**Connection of** (13) **to a Convex Program.** Let $Z = HW \in \mathbb{R}^{K \times N}$ with $N = nK$ and $\alpha = \frac{\lambda_H}{\lambda_W}$. By Lemma C.3, we know that

$$\min_{HW=Z} \frac{\lambda_W}{2} \|W\|_F^2 + \frac{\lambda_H}{2} \|H\|_F^2 = \sqrt{\lambda_W \lambda_H} \min_{HW=Z} \frac{1}{2\sqrt{\alpha}} \left( \|W\|_F^2 + \alpha \|H\|_F^2 \right)$$
$$= \sqrt{\lambda_W \lambda_H} \|Z\|_* .$$

Thus, we can relate the bilinear factorized problem (13) to a convex problem in terms of $Z$ and $b$ as follows

$$\min_{Z \in \mathbb{R}^{K \times N}, \, b \in \mathbb{R}^K} \widetilde{f}(Z, b) := g\left(Z + b\mathbf{1}^\top\right) + \sqrt{\lambda_W \lambda_H} \|Z\|_* + \frac{\lambda_b}{2} \|b\|_2^2. \qquad (36)$$

Similar to the idea in [32,77,87,115], we will characterize the critical points of (13) by establishing a connection to the optimality condition of the convex problem (36). Towards this goal, we first show the global minimum of the convex program (36) provides a lower bound for the original problem (3). More specifically, in Lemma E.2 we can show that for any global minimizer $(Z_\star, b_\star)$ of $\widetilde{f}$ satisfies

$$\widetilde{f}(Z^\star, b^\star) \leq f(W, H, b), \quad \forall \, W \in \mathbb{R}^{K \times d}, \, H \in \mathbb{R}^{d \times N}, \, b \in \mathbb{R}^K. \qquad (37)$$

**Characterizing the Optimality Condition of** (13) **Based on the Convex Program** (36). Next, we characterize the optimality condition of our nonconvex problem (13), based on the relationship to its convex counterpart (36). Specifically, Lemma E.3 showed that any critical point $(Z, b)$ of (36) is characterized by the following necessary and sufficient condition

$$\nabla g(Z + b\mathbf{1}^\top) \in -\sqrt{\lambda_W \lambda_H} \partial \|Z\|_*,$$
$$\sum_{i=1}^N [\nabla g(Z + b\mathbf{1}^\top)]_i + \lambda_b b = 0, \qquad (38)$$

where $[\nabla g(Z + b\mathbf{1}^\top)]_i$ represents the $i$-th column in $\nabla g(Z + b\mathbf{1}^\top)$. By Lemma E.4, we can transfer the optimality condition from convex to the nonconvex problem (13). More specifically, any critical point $(W, H, b)$ of (13) satisfies

$$\left\| \nabla g(WH + b\mathbf{1}^\top) \right\| \leq \sqrt{\lambda_W \lambda_H},$$

then $(Z, b)$ with $Z = WH$ satisfies all the conditions in (38). Combining with (37), Lemma E.4 showed that $(W, H, b)$ is a global solution of the nonconvex problem (13).

**Proving No Spurious Local Minima and Strict Saddle Property.** Now we turn to prove the strict saddle property and that there are no spurious local minima by characterizing the properties for all the critical points of (13). Denote the set of critical points of $f(W, H, b)$ by

$$\mathcal{C} := \left\{ W \in \mathbb{R}^{K \times d}, H \in \mathbb{R}^{d \times N}, b \in \mathbb{R}^K \mid \nabla f(W, H, b) = 0 \right\}.$$

To proceed, we separate the set $\mathcal{C}$ into two disjoint subsets

$$\mathcal{C}_1 := \mathcal{C} \cap \left\{ W \in \mathbb{R}^{K \times d}, H \in \mathbb{R}^{d \times N}, b \in \mathbb{R}^K \mid \left\| \nabla g(WH + b\mathbf{1}^\top) \right\| \leq \sqrt{\lambda_W \lambda_H} \right\},$$
$$\mathcal{C}_2 := \mathcal{C} \cap \left\{ W \in \mathbb{R}^{K \times d}, H \in \mathbb{R}^{d \times N}, b \in \mathbb{R}^K \mid \left\| \nabla g(WH + b\mathbf{1}^\top) \right\| > \sqrt{\lambda_W \lambda_H} \right\},$$

satisfying $\mathcal{C} = \mathcal{C}_1 \cup \mathcal{C}_2$. By Lemma E.4, we already know that any $(W, H, b) \in \mathcal{C}_1$ is a global optimal solution of $f(W, H, b)$ in (13). If we can show that any critical point in $\mathcal{C}_2$ possesses negative curvatures (i.e., the Hessian at $(W, H, b)$ has at least one negative eigenvalue), then we prove that there is no spurious local minima as well as strict saddle property.

Thus, the remaining part is to show any point in $\mathcal{C}_2$ possesses negative curvatures. We will find a direction $\Delta$ along which the Hessian has a strictly negative curvature for this point. Towards that goal, for any $\Delta = (\Delta_W, \Delta_H, \Delta_b)$, we first compute the Hessian bilinear form of $f(W, H, b)$ along the direction $\Delta$ by

$$\nabla^2 f(W, H, b)[\Delta, \Delta]$$
$$= \nabla^2 g(WH + b\mathbf{1}^\top) \left[ W\Delta_H + \Delta_W H + \Delta_b \mathbf{1}^\top, W\Delta_H + \Delta_W H + \Delta_b \mathbf{1}^\top \right] \qquad (39)$$
$$+ 2 \left\langle \nabla g(WH + b\mathbf{1}^\top), \Delta_W \Delta_H \right\rangle + \lambda_W \|\Delta_W\|_F^2 + \lambda_H \|\Delta_H\|_F^2 + \lambda_b \|\Delta_b\|_2^2.$$

We now utilize the property that $\left\|\nabla g(\boldsymbol{W}\boldsymbol{H} + \boldsymbol{b}\mathbf{1}^\top)\right\| > \sqrt{\lambda_{\boldsymbol{W}}\lambda_{\boldsymbol{H}}}$ for any $(\boldsymbol{W}, \boldsymbol{H}, \boldsymbol{b}) \in \mathcal{C}_2$ to construct a negative curvature direction. Let $\boldsymbol{u}$ and $\boldsymbol{v}$ be the left and right singular vectors corresponding to the largest singular value $\sigma_1(\nabla^2 g(\boldsymbol{W}\boldsymbol{H} + \boldsymbol{b}\mathbf{1}^\top))$ of $\nabla^2 g(\boldsymbol{W}\boldsymbol{H} + \boldsymbol{b}\mathbf{1}^\top)$, which is larger than $\sqrt{\lambda_{\boldsymbol{W}}\lambda_{\boldsymbol{H}}}$ by our assumption.

Since $d > K$ and $\boldsymbol{W} \in \mathbb{R}^{K \times d}$, there exists a nonzero $\boldsymbol{a} \in \mathbb{R}^d$ in the null space of $\boldsymbol{W}$, i.e., $\boldsymbol{W}\boldsymbol{a} = \boldsymbol{0}$. Furthermore, by Lemma D.2, we know that

$$\boldsymbol{W}^\top\boldsymbol{W} = \sqrt{\frac{\lambda_{\boldsymbol{H}}}{\lambda_{\boldsymbol{W}}}}\boldsymbol{H}\boldsymbol{H}^\top \quad \Longrightarrow \quad \boldsymbol{H}^\top\boldsymbol{a} = \boldsymbol{0}.$$

We now construct the negative curvature direction as

$$\boldsymbol{\Delta} = (\boldsymbol{\Delta}_{\boldsymbol{W}}, \boldsymbol{\Delta}_{\boldsymbol{H}}, \boldsymbol{\Delta}_{\boldsymbol{b}}) = \left( \left(\frac{\lambda_{\boldsymbol{H}}}{\lambda_{\boldsymbol{W}}}\right)^{1/4} \boldsymbol{u}\boldsymbol{a}^\top, -\left(\frac{\lambda_{\boldsymbol{H}}}{\lambda_{\boldsymbol{W}}}\right)^{-1/4} \boldsymbol{a}\boldsymbol{v}^\top, \boldsymbol{0} \right)$$

so that the term $\langle \nabla g(\boldsymbol{W}\boldsymbol{H} + \boldsymbol{b}\mathbf{1}^\top), \boldsymbol{\Delta}_{\boldsymbol{W}}\boldsymbol{\Delta}_{\boldsymbol{H}} \rangle$ is small enough to create a negative curvature. Since $\boldsymbol{W}\boldsymbol{a} = \boldsymbol{0}$, $\boldsymbol{a}^\top\boldsymbol{H} = \boldsymbol{0}$, and $\boldsymbol{\Delta}_{\boldsymbol{b}} = \boldsymbol{0}$, we have

$$\boldsymbol{W}\boldsymbol{\Delta}_{\boldsymbol{H}} + \boldsymbol{\Delta}_{\boldsymbol{W}}\boldsymbol{H} + \boldsymbol{\Delta}_{\boldsymbol{b}}\mathbf{1}^\top = -\left(\frac{\lambda_{\boldsymbol{H}}}{\lambda_{\boldsymbol{W}}}\right)^{-1/4} \boldsymbol{W}\boldsymbol{a}\boldsymbol{v}^\top + \left(\frac{\lambda_{\boldsymbol{H}}}{\lambda_{\boldsymbol{W}}}\right)^{1/4} \boldsymbol{u}\boldsymbol{a}^\top\boldsymbol{H} = \boldsymbol{0},$$

so that $\nabla^2 g(\boldsymbol{W}\boldsymbol{H} + \boldsymbol{b}\mathbf{1}^\top)\left[\boldsymbol{W}\boldsymbol{\Delta}_{\boldsymbol{H}} + \boldsymbol{\Delta}_{\boldsymbol{W}}\boldsymbol{H} + \boldsymbol{\Delta}_{\boldsymbol{b}}\mathbf{1}^\top, \boldsymbol{W}\boldsymbol{\Delta}_{\boldsymbol{H}} + \boldsymbol{\Delta}_{\boldsymbol{W}}\boldsymbol{H} + \boldsymbol{\Delta}_{\boldsymbol{b}}\mathbf{1}^\top\right] = 0$. Thus, combining the results above with (39), we obtain the following

$$\begin{aligned} &\nabla f(\boldsymbol{W}, \boldsymbol{H}, \boldsymbol{b})[\boldsymbol{\Delta}, \boldsymbol{\Delta}] \\ &= 2\left\langle \nabla g(\boldsymbol{W}\boldsymbol{H} + \boldsymbol{b}\mathbf{1}^\top), \boldsymbol{\Delta}_{\boldsymbol{W}}\boldsymbol{\Delta}_{\boldsymbol{H}} \right\rangle + \lambda_{\boldsymbol{W}}\left\|\boldsymbol{\Delta}_{\boldsymbol{W}}\right\|_F^2 + \lambda_{\boldsymbol{H}}\left\|\boldsymbol{\Delta}_{\boldsymbol{H}}\right\|_F^2 \\ &= -2\left\|\boldsymbol{a}\right\|_2^2 \left\langle \nabla g(\boldsymbol{W}\boldsymbol{H} + \boldsymbol{b}\mathbf{1}^\top), \boldsymbol{u}\boldsymbol{v}^\top \right\rangle + 2\sqrt{\lambda_{\boldsymbol{W}}\lambda_{\boldsymbol{H}}}\left\|\boldsymbol{a}\right\|_2^2 \\ &= -2\left\|\boldsymbol{a}\right\|_2^2 \left(\left\|\nabla g(\boldsymbol{W}\boldsymbol{H} + \boldsymbol{b}\mathbf{1}^\top)\right\| - \sqrt{\lambda_{\boldsymbol{W}}\lambda_{\boldsymbol{H}}}\right) < 0, \end{aligned}$$

where the last inequality is based on the fact that $(\boldsymbol{W}, \boldsymbol{H}, \boldsymbol{b}) \in \mathcal{C}_2$ so that $\left\|\nabla g(\boldsymbol{W}\boldsymbol{H} + \boldsymbol{b}\mathbf{1}^\top)\right\| > \sqrt{\lambda_{\boldsymbol{W}}\lambda_{\boldsymbol{H}}}$. Therefore, any $(\boldsymbol{W}, \boldsymbol{H}, \boldsymbol{b}) \in \mathcal{C}_2$ possesses at least one negative curvature direction. This completes our proof of Theorem E.1. ■

### E.2 Supporting Lemmas

**Lemma E.2** *If $(\boldsymbol{Z}^\star, \boldsymbol{b}^\star)$ is a global minimizer of*

$$\min_{\boldsymbol{Z} \in \mathbb{R}^{K \times N}, \, \boldsymbol{b} \in \mathbb{R}^K} \widetilde{f}(\boldsymbol{Z}, \boldsymbol{b}) := g\left(\boldsymbol{Z} + \boldsymbol{b}\mathbf{1}^\top\right) + \sqrt{\lambda_{\boldsymbol{W}}\lambda_{\boldsymbol{H}}}\left\|\boldsymbol{Z}\right\|_* + \frac{\lambda_{\boldsymbol{b}}}{2}\left\|\boldsymbol{b}\right\|_2^2.$$

*introduced in (36), then $\widetilde{f}(\boldsymbol{Z}^\star, \boldsymbol{b}^\star) \leq f(\boldsymbol{W}, \boldsymbol{H}, \boldsymbol{b})$ for all $\boldsymbol{W} \in \mathbb{R}^{K \times d}, \boldsymbol{H} \in \mathbb{R}^{d \times N}, \boldsymbol{b} \in \mathbb{R}^K$.*

**Proof** [Proof of Lemma E.2] Suppose $(\boldsymbol{Z}^\star, \boldsymbol{b}^\star)$ is a global minimum of $\widetilde{f}(\boldsymbol{Z}, \boldsymbol{b})$. Then by Theorem C.3, we have

$$\begin{aligned} \widetilde{f}(\boldsymbol{Z}^\star, \boldsymbol{b}^\star) &= \min_{\boldsymbol{Z}, \boldsymbol{b}} g(\boldsymbol{Z} + \boldsymbol{b}\mathbf{1}^\top) + \sqrt{\lambda_{\boldsymbol{W}}\lambda_{\boldsymbol{H}}}\left\|\boldsymbol{Z}\right\|_* + \frac{\lambda_{\boldsymbol{b}}}{2}\left\|\boldsymbol{b}\right\|_2^2 \\ &= \min_{\boldsymbol{Z}, \boldsymbol{b}} g(\boldsymbol{Z} + \boldsymbol{b}\mathbf{1}^\top) + \min_{\boldsymbol{W}\boldsymbol{H} = \boldsymbol{Z}} \left(\frac{\lambda_{\boldsymbol{W}}}{2}\left\|\boldsymbol{W}\right\|_F^2 + \frac{\lambda_{\boldsymbol{H}}}{2}\left\|\boldsymbol{H}\right\|_F^2\right) + \frac{\lambda_{\boldsymbol{b}}}{2}\left\|\boldsymbol{b}\right\|_2^2 \\ &\leq \min_{\boldsymbol{W}, \boldsymbol{H}, \boldsymbol{Z}, \boldsymbol{b}, \boldsymbol{Z} = \boldsymbol{W}\boldsymbol{H}} g(\boldsymbol{W}\boldsymbol{H} + \boldsymbol{b}\mathbf{1}^\top) + \frac{\lambda_{\boldsymbol{W}}}{2}\left\|\boldsymbol{W}\right\|_F^2 + \frac{\lambda_{\boldsymbol{H}}}{2}\left\|\boldsymbol{H}\right\|_F^2 + \frac{\lambda_{\boldsymbol{b}}}{2}\left\|\boldsymbol{b}\right\|_2^2. \end{aligned}$$

Thus, we have

$$\widetilde{f}(\boldsymbol{Z}^\star, \boldsymbol{b}^\star) \leq \min_{\boldsymbol{W} \in \mathbb{R}^{K \times d}, \boldsymbol{H} \in \mathbb{R}^{d \times N}, \boldsymbol{b} \in \mathbb{R}^K} f(\boldsymbol{W}, \boldsymbol{H}, \boldsymbol{b})$$

as desired. ■

**Lemma E.3 (Optimality Condition for the Convex Program** (36)**)** *Consider the following convex program in* (36) *that we rewrite as follows*

$$\min_{\boldsymbol{Z} \in \mathbb{R}^{K \times N}, \, \boldsymbol{b} \in \mathbb{R}^K} \widetilde{f}(\boldsymbol{Z}, \boldsymbol{b}) \; := \; g\left(\boldsymbol{Z} + \boldsymbol{b}\boldsymbol{1}^\top\right) + \sqrt{\lambda_W \lambda_H} \, \|\boldsymbol{Z}\|_* + \frac{\lambda_b}{2} \|\boldsymbol{b}\|_2^2.$$

*Then the necessary and sufficient condition for* $(\boldsymbol{Z}, \boldsymbol{b})$ *being the global solution of* (36) *is*

$$\nabla g(\boldsymbol{Z} + \boldsymbol{b}\boldsymbol{1}^\top)\boldsymbol{V} \; = \; -\sqrt{\lambda_W \lambda_H}\boldsymbol{U}, \quad \nabla g(\boldsymbol{Z} + \boldsymbol{b}\boldsymbol{1}^\top)^\top \boldsymbol{U} \; = \; -\sqrt{\lambda_W \lambda_H}\boldsymbol{V},$$

$$\left\|\nabla g(\boldsymbol{Z} + \boldsymbol{b}\boldsymbol{1}^\top)\right\| \; \leq \; \sqrt{\lambda_W \lambda_H}, \quad and \quad \sum_{i=1}^N [\nabla g(\boldsymbol{Z} + \boldsymbol{b}\boldsymbol{1}^\top)]_i + \lambda_b \boldsymbol{b} \; = \; \boldsymbol{0}, \tag{40}$$

*where* $\boldsymbol{U}$ *and* $\boldsymbol{V}$ *are the left and right singular value matrices of* $\boldsymbol{Z}$*, i.e.,* $\boldsymbol{Z} = \boldsymbol{U}\boldsymbol{\Sigma}\boldsymbol{V}^\top$.

**Proof** [Proof of Lemma E.3] Standard convex optimization theory asserts that any critical point $(\boldsymbol{Z}, \boldsymbol{b})$ of (36) is global, where the optimality is characterized by the following necessary and sufficient condition

$$\nabla g(\boldsymbol{Z} + \boldsymbol{b}\boldsymbol{1}^\top) \; \in \; -\sqrt{\lambda_W \lambda_H} \partial \|\boldsymbol{Z}\|_*,$$

$$\sum_{i=1}^N [\nabla g(\boldsymbol{Z} + \boldsymbol{b}\boldsymbol{1}^\top)]_i + \lambda_b \boldsymbol{b} \; = \; \boldsymbol{0}, \tag{41}$$

where $[\nabla g(\boldsymbol{Z} + \boldsymbol{b}\boldsymbol{1}^\top)]_i$ represents the $i$-th column in $\nabla g(\boldsymbol{Z} + \boldsymbol{b}\boldsymbol{1}^\top)$, and $\partial \|\boldsymbol{Z}\|_*$ denotes the subdifferential of the convex nuclear norm $\|\boldsymbol{Z}\|_*$ evaluated at $\boldsymbol{Z}$. By its definition, we have

$$\partial \|\boldsymbol{Z}\|_* \; := \; \left\{ \boldsymbol{D} \in \mathbb{R}^{K \times N} \; \mid \; \|\boldsymbol{G}\|_* \geq \|\boldsymbol{Z}\|_* + \langle \boldsymbol{G} - \boldsymbol{Z}, \boldsymbol{D} \rangle, \; \boldsymbol{G} \in \mathbb{R}^{K \times N} \right\},$$

where for nuclear norm, the previous work [116, 117] showed that based on the projection onto the column space and row space via the SVD of $\boldsymbol{Z} = \boldsymbol{U}\boldsymbol{\Sigma}\boldsymbol{V}^\top$, this is equivalent to

$$\partial \|\boldsymbol{Z}\|_* \; = \; \left\{ \boldsymbol{U}\boldsymbol{V}^\top + \boldsymbol{W}, \boldsymbol{W} \in \mathbb{R}^{K \times N} \; \mid \; \boldsymbol{U}^\top \boldsymbol{W} = \boldsymbol{0}, \; \boldsymbol{W}\boldsymbol{V} = \boldsymbol{0}, \; \|\boldsymbol{W}\| \leq 1 \right\},$$

where $\boldsymbol{U}$ and $\boldsymbol{V}$ are the left and right singular value matrices of $\boldsymbol{Z}$. Using the result above, we can now rewrite the optimality condition in (41) as suggested in Lemma E.3. ∎

**Lemma E.4 (Optimality Condition for the Nonconvex Program** (13)**)** *If a critical point* $(\boldsymbol{W}, \boldsymbol{H}, \boldsymbol{b})$ *of* (13) *satisfies*

$$\left\|\nabla g(\boldsymbol{W}\boldsymbol{H} + \boldsymbol{b}\boldsymbol{1}^\top)\right\| \; \leq \; \sqrt{\lambda_W \lambda_H}, \tag{42}$$

*then it is a global minimum of* (13).

**Proof** [Proof of Lemma E.4] Suppose $(\boldsymbol{W}^\star, \boldsymbol{H}^\star, \boldsymbol{b}^\star)$ is a critical point of (13) satisfying (42), we will show that $(\boldsymbol{Z}^\star, \boldsymbol{b}^\star)$ with $\boldsymbol{Z}^\star = \boldsymbol{W}^\star \boldsymbol{H}^\star$ is a global minimizer of (36) by showing that $(\boldsymbol{W}^\star \boldsymbol{H}^\star, \boldsymbol{b}^\star)$ satisfies the optimality condition for the convex program in (40) (Lemma E.3). First of all, it is easy to check that

$$\nabla_{\boldsymbol{b}} f(\boldsymbol{W}^\star, \boldsymbol{H}^\star, \boldsymbol{b}^\star) \; = \; \sum_{i=1}^N [\nabla g(\boldsymbol{W}^\star \boldsymbol{H}^\star + \boldsymbol{b}^\star \boldsymbol{1}^\top)]_i + \lambda_b \boldsymbol{b}^\star \; = \; \boldsymbol{0}$$

$$\implies \sum_{i=1}^N [\nabla g(\boldsymbol{Z}^\star + \boldsymbol{b}^\star \boldsymbol{1}^\top)]_i + \lambda_b \boldsymbol{b}^\star \; = \; \boldsymbol{0}.$$

Second, let $\boldsymbol{Z}^\star = \boldsymbol{W}^\star \boldsymbol{H}^\star = \boldsymbol{U}\boldsymbol{\Sigma}\boldsymbol{V}^\top$ be the compact SVD of $\boldsymbol{Z}^\star = \boldsymbol{W}^\star \boldsymbol{H}^\star$. By Lemma D.2, we have

$$\boldsymbol{W}^{\star\top}\boldsymbol{W}^\star \; = \; \frac{\lambda_H}{\lambda_W} \boldsymbol{H}^\star \boldsymbol{H}^{\star\top} \implies \boldsymbol{H}^{\star\top}\boldsymbol{H}^\star \boldsymbol{H}^{\star\top}\boldsymbol{H}^\star \; = \; \frac{\lambda_W}{\lambda_H} \boldsymbol{H}^{\star\top}\boldsymbol{W}^{\star\top}\boldsymbol{W}^\star \boldsymbol{H}^\star \; = \; \frac{\lambda_W}{\lambda_H} \boldsymbol{V}\boldsymbol{\Sigma}^2 \boldsymbol{V}^\top.$$

Now by utilizing Lemma C.4, from the above equation we obtain the following

$$\boldsymbol{H}^{\star\top}\boldsymbol{H}^\star \; = \; \sqrt{\frac{\lambda_W}{\lambda_H}} \boldsymbol{V}\boldsymbol{\Sigma}\boldsymbol{V}^\top.$$

This together with the first equation in (16) gives

$$\nabla g(\boldsymbol{W}^\star \boldsymbol{H}^\star + \boldsymbol{b}^\star \mathbf{1}^\top) \boldsymbol{H}^{\star\top} = -\lambda_{\boldsymbol{W}} \boldsymbol{W}^\star \implies \nabla g(\boldsymbol{W}^\star \boldsymbol{H}^\star + \boldsymbol{b}^\star \mathbf{1}^\top) \boldsymbol{H}^{\star\top} \boldsymbol{H}^\star = -\lambda_{\boldsymbol{W}} \boldsymbol{W}^\star \boldsymbol{H}^\star$$
$$\implies \nabla g(\boldsymbol{W}^\star \boldsymbol{H}^\star + \boldsymbol{b}\mathbf{1}^\top) \sqrt{\frac{\lambda_{\boldsymbol{W}}}{\lambda_{\boldsymbol{H}}}} \boldsymbol{V} \boldsymbol{\Sigma} \boldsymbol{V}^\top = -\lambda_{\boldsymbol{W}} \boldsymbol{U} \boldsymbol{\Sigma} \boldsymbol{V}^\top$$
$$\implies \nabla g(\boldsymbol{Z}^\star + \boldsymbol{b}^\star \mathbf{1}^\top) \boldsymbol{V} = -\sqrt{\lambda_{\boldsymbol{W}} \lambda_{\boldsymbol{H}}} \boldsymbol{U}.$$

Similarly, we can also get

$$\nabla g(\boldsymbol{Z}^\star + \boldsymbol{b}^\star \mathbf{1}^\top)^\top \boldsymbol{U} = -\sqrt{\lambda_{\boldsymbol{W}} \lambda_{\boldsymbol{H}}} \boldsymbol{V}.$$

Thus, together with (42), $(\boldsymbol{Z}^\star, \boldsymbol{b}^\star)$ with $\boldsymbol{Z}^\star = \boldsymbol{W}^\star \boldsymbol{H}^\star$ satisfies the optimality condition (40), and hence is a global minimizer of $\widetilde{f}(\boldsymbol{Z}^\star, \boldsymbol{b}^\star)$ in (36).

Finally, we complete the proof by invoking Lemma E.2. By Lemma E.2 with $\boldsymbol{Z}^\star = \boldsymbol{W}^\star \boldsymbol{H}^\star$, we know that $f(\boldsymbol{W}^\star, \boldsymbol{H}^\star, \boldsymbol{b}^\star) = \widetilde{f}(\boldsymbol{Z}^\star, \boldsymbol{b}^\star) \le f(\boldsymbol{W}, \boldsymbol{H}, \boldsymbol{b})$ for all $\boldsymbol{W} \in \mathbb{R}^{K \times d}, \boldsymbol{H} \in \mathbb{R}^{d \times N}, \boldsymbol{b} \in \mathbb{R}^K$. Therefore, we must have $(\boldsymbol{W}^\star, \boldsymbol{H}^\star, \boldsymbol{b}^\star)$ to be the global solution of $f(\boldsymbol{W}, \boldsymbol{H}, \boldsymbol{b})$ in (13). ∎