# OpenReview forum: "A Geometric Analysis of Neural Collapse with Unconstrained Features"
_NeurIPS.cc/2021/Conference — NeurIPS 2021 Spotlight_

### Official Review · Reviewer_7gsu · 2021-07-09

**Rating:** 7
**Confidence:** 5

**Summary:**

This paper provides a landscape analysis for neural collapse with regularization analysis. The theoretical results are nice, thus I voted an accept decision. But reviewer has a concern about the novelty of the proof and the implications of the theoretical results.

**Limitations And Societal Impact:**

see detailed review

**Main Review:**

this is a nice paper analysis the landscape of the neural collapse model, The main concern reviewer have are following
[1] The novelty of no strictly bad local minimum:
The analysis is actually the same as the landscape analysis of the matrix decomposition model in my mind. The paper deal with the problem as a convex_function(WH)+\lambda ||W||^2_F +\lambda ||H||^2_F where the nonconvexity is introduced by WH. I wonder whether the landscape results are the same as the previous matrix decomposition paper

[2] The novelty of the proves of the unconstrained model. I wonder can we use the Lagrange dual relationship turns the global minimum of the unconstrained one to a constrained one. In practice, no one use regularization on features, I still can't see the benefit and novelty to turn the constrained model to a unconstrained one.

[3] The implications of the theoretical results are separated from the theoretical results in the paper. The "new method" you can do that even after the first neural collapse paper and the "new" methods can also be understood as changing a loss function[using target feature as loss] and this leads to question 5.

[4] From the theoretical result, can we say the layer before the last layer is also neural collapse? I wonder will the proof stays true while we inductive on the depth.

**Time Spent Reviewing:**

1.5

---

> ### Author Response · Authors · 2021-08-10
> **Response to Reviewer 7gsu**
>
> We thank the reviewer for the detailed and thoughtful comments. In the following, we address the reviewer’s comments one by one.
>
> ## Interpretation of our results & relationship to low-rank matrix recovery
>
> Indeed, our geometric analysis of critical points for the nonconvex loss function (3) draws connections to recent advances on nonconvex low-rank matrix recovery problems, where we showed similar strict saddle properties with analogous analytical tools. However, we believe the inherent connections between our problem and nonconvex low-rank matrix recovery is a strength of our work instead of a weakness. We believe the contribution of our work is more at a conceptual level rather than at a technical level. This is the first result showing that the mathematical tools developed for low-rank matrix factorization can provide systematic approaches for studying interesting empirically justified phenomena in deep network training. Our work connects the two research communities of low-rank matrix factorization with deep learning, opening and pointing to new promising directions for further investigation.
>
> That being said, however, there are several fundamental differences between the two problems of low-rank matrix factorization and deep network training:
>
> * **Differences in tasks: recovery vs training classifiers**. For low-rank matrix factorization, the task is often posed as a recovery problem, whose goal is to recover a ground-truth low-rank matrix. Thus, those works are focused on the study of recovery conditions (e.g., sample complexity, statistical properties of the sensing matrices). In comparison, here we are studying neural network training for a multi-class classification problem which is not a recovery problem. Thus, the output $\bf y$ are structured as one-hot label vectors, and we are instead interested in the properties of the learned features $\bf H$, classifiers $\bf W$, and their relationships.
>
> * **Differences in global solutions**. Because of the difference in tasks, global solutions are different. For low-rank matrix recovery, the global optimal solutions are the ground-truth low-rank matrices up to rotations (assumed given in analysis). For our problem, the global solutions for the features and classifiers are simplex ETFs with neural collapse.
>
> * **Differences in losses and statistical properties**. Because of the differences in tasks, the loss functions are different: we analyze the classification problem under the commonly used cross-entropy loss, while for recovery problems it typically uses least-squares loss. The cross-entropy loss introduces extra technical difficulties in the analysis. On the other hand, for low-rank matrix recovery problems, people often study the problem under certain randomness or statistical properties of the sensing matrices. In contrast, these statistical properties do not exist in our problem, where the model and analysis are purely deterministic.
>
> Some of the similarities and differences have been discussed in Section 3.1 (page 6, lines 232-239). In the revision, we will discuss these in more detail following the reviewer’s suggestions.
>
> ## Unconstrained problem formulation vs. the constrained counterpart
> In the current practice of deep learning for classification problems, the most commonly used loss function is cross-entropy with weight decay on the network parameters (e.g., adding an $\ell^2$-regularization on all the network weights $\bf \Theta$). In comparison, our training loss formulation (3) is enforcing the $\ell^2$-regularization on the features $\bf H$ rather than on the network parameters $\bf \Theta$. However, it should be noted the following:
>
> * **Similarity with common practices**: Compared to the constrained formulation on the features, our unconstrained formulation (i.e., enforcing the $\ell_2$-regularization on the feature $\bf H$) appears to be closer to the practically used unconstrained loss function with weight decay;
> * **Experimental verification**: Our experimental results in Appendix B.2.2 (i.e., line 803-807) and Figure 9 demonstrate that our formulation has similar performance with commonly used weight decay formulation, in terms of neural collapse and generalization performance. This implies that our idealization is reasonable.
>
> In the paper, we have discussed the validity of our loss in Section 2.1 (page 4, line 151 - 157) and experimentally verified in Appendix B.2.2. We will make this more clear in the revision.
>
> ## Lagrange dual relationship turns the global minimum of the unconstrained one to a constrained one.
> The reviewer has raised a very good point. By Lagrange duality, it is possible to turn the global minimum of the unconstrained model into a constrained one. But whether the benign global landscape will be preserved is unclear. We conjecture this is also true (i.e., the constrained one also has a benign global landscape) considering the special form of the constraint, i.e., $\ell_2$ norm. We think this is a good extension of our work and will leave the formal justification as future work.
>
> ## Implication of our theoretical results for practice.
> It should be noted that the first neural collapse paper is an empirical work, while our results have provided rigorous theoretical explanations for why the neural collapse happens in practical network training via characterizing all the critical points (under the assumption of the unconstrained feature model). Thus, the new network design in Section 4.3 is based on theoretical principles rather than pure empirical evidence, and it is directly suggested by our new analytical results. Moreover, our theoretical results have provided new perspectives on practical network training, which cannot be obtained or has been neglected in the first neural collapse work:
>
> * **Neural collapse is algorithmic independent**. Because of our global landscape analysis, we showed that neural collapse is algorithmic independent and numerically verified through experiments (see Section 4.1).
> * **Dimension reduction for the feature and classifier**. Our theoretical analysis allows us to pinpoint $d = K$ (i.e., feature dimension = number of classes) as the critical threshold for the feature dimension and assert that neural collapse occurs if $d \ge K$. This provides the theoretical justification for reducing feature dimension to $d = K$ which as we numerically verify does not hurt training and generalization performance (see Sec. 4.3, line 344-354).
>
> ## Extension of studying features before the last layer inductively
> This is a good suggestion for the extension of our current results. Actually, there are some recent experimental results in [16] (Figure 8), showing that features in shallower layers are progressively exhibiting less collapse. Extension of our current study faces two challenges:
>
> * *Dealing with nonlinear activation*. The nonlinearity activation such as ReLU is adding extra nonsmooth and nonconvexity in the analysis.
> * *Universal approximation*. Also note that the unconstrained feature model relies on the assumption that the feature mapping is a universal approximator, which may not hold for the shallower layers.
>
> Nonetheless, the recent work [A] studied a three-layer random feature model with nonlinear activation, where they showed a global optimality condition that is similar to neural collapse. These results [16, A] might inspire interesting ideas/directions for future investigation, which we will add discussion in the conclusion part of the paper.
>
> ### References mentioned
>
> [16] Vardan Papyan. Traces of class/cross-class structure pervade deep learning spectra. Journal of Machine Learning Research, 21(252):1–64, 2020.
>
> [A] Nishanth Dikkala, Gal Kaplun, Rina Panigrahy, For Manifold Learning, Deep Neural Networks can be Locality Sensitive Hash Functions, arXiv preprint arXiv:2103.06875

---

> > ### Comment · Reviewer_7gsu · 2021-08-11
> > **###**
> >
> > Thanks the reviewer for answering the questions. They answered most of my concern and I would like to see some of the discussion added to the main draft.(mostly relationship to low-rank matrix recovery)
> >
> > I decide to raise the core.(6->7)

---

> > > ### Author Response · Authors · 2021-08-11
> > > **Reply to Reviewer 7gsu**
> > >
> > > We thank the reviewer for the appreciation of our efforts in the response. We will incorporate these discussions in the revision.

---

### Official Review · Reviewer_HfZN · 2021-07-16

**Rating:** 7
**Confidence:** 4

**Summary:**

The paper studies collapse phenomena of representations from neural classification models trained beyond zero training error with cross entropy. The theoretical contribution of the paper is an analysis of such phenomena in the so called unconstrained feature model. In this setting, the authors derive the global loss minimizers, and show that these can be obtained via gradient based optimization, as the critical points of the loss function are benign. These theoretical results are underpinned by experiments, which investigate the collapse phenomena for a ResNet-18 architecture when trained via different optimization algorithms or with randomly labeled data.
The theoretical results also indicate, that in practice, the last linear layer of a neural network can be fixed a priory and of low dimension.


**Limitations And Societal Impact:**

I agree with the authors, that there is no negative societal impact to be expected from this work. As acknowledged by the authors, the work is limited to the unconstrained feature model.

**Main Review:**


## Overall assessment:

This is an interesting paper which promotes the understanding of loss minimizers and loss optimization in the supervised classification setting. There are several issues regarding the assumptions and claims made in the paper, but I think these can be cleared with a revision. Furthermore, its connection to related work is described kind of superficially in the paper. In general, this is a good submission and I am willing to raise my score, once the major issues are resolved.



### Significance/Originality:

Inspired by the observed neural collapse phenomenon in Papyan et al, there is a recent stream of research, which studies such phenomena from a theoretical perspective. Mainly, by deriving the loss minimizers under various assumptions in the unconstrained feature model. The first theoretical result in this paper is quite similar to these works.
The second theoretical result provides a real addition to the literature. There, in contrast to previous work, the authors do analyze the critical points in the unconstrained feature model and thus take into account that neural networks are usually trained via gradient based optimization.
In summary, the paper promotes our understanding of the neural collapse phenomenon.
---

While previous work relies on norm constraints on the representations, the authors argue, that weight decay yields an implicit L2-penalty on the representations. This is a quite interesting and original assumption. The authors argue that the assumption is meaningful, but unfortunately, this was not discussed more in detail, see Quality.

---
From the paper itself, it is difficult for readers to assess its role in the literature. This is because the authors tend to simultaneously cite multiple references which are only summarized as a whole. As a consequence, most references and how they relate to the submitted work are not discussed properly, leading to superficial related work sections. In the following paragraphs, I will precise this with some examples from the text.

For example, the section related to existing landscape analysis consists mainly of the statement that the authors' "result has much broader implications than the previous results [31, 32, 35, 36, 38, 39, 85]" (line 244), however the contribution of these earlier works, and thus the reason for the, (according to the authors) limited implications of these works remains unclear. Addendum: There is a section in the supplementary material which discusses these works in more detail. The existence of this section and its most important parts should be highlighted in the main text.

The relationship to low rank-matrix recovery section suffers from similar problems. The authors write "As discussed in Section 1, it has been recently shown that the strict saddle property holds for a wide range of nonconvex problems in machine learning [66–79], including low-rank matrix recovery [74, 76, 80–83]" (line 233). (I did not find this in section 1, maybe the authors could specify the exact location of the discussion in section 1.) \
First of all, I don't think that there is much benefit in simply listing machine learning papers which study critical points of some functions. The authors should either expand on the relationship of these references to the submitted work, or remove them.\
Furthermore, this list ([66–79]) is far from complete.  To name a few additional references, critical points were also studied for residual networks, see [5'] and [6']. There is also the work by Kawaguchi, which the authors already cite (reference [32]), but not in this context. \
In fact, judging from the titles of [66–79], this list seems limited to the subarea of low rank matrix optimization and is due to mainly the same group of authors. If the point of including these references is to illustrate the existence of strict saddle points in many machine learning problems, then it is counterproductive to limit the references to a particular subfield. But as already stated, the main problem is not the lack of the references, but the superficial treatment of the provided ones.\
I am aware, that the works [66–79] are also presented in the supplementary material (starting at line 725). But even there, only the application areas of these works are named, but how these works relate to each other and the submitted work is not discussed. For example, it remains unclear, whether there is a central paper which introduced the required mathematics and the others are just applications of it, or if these works all build on each other and, if yes, how and to which extend. At the end of this paragraph in the supplementary material, two papers are highlighted; "In fact, our proof techniques are inspired by recent results on low-rank matrix recovery [73, 76]". But again, what these results entail, and how they inspired the authors' proof techniques remains unspecified.
---

I also want to highlight some related work not mentioned in the paper.

Assuming that the representations and classifier weights are constrained on a sphere and discarding bias terms, Wang et al.[1'] already showed in 2017 that if the representations are already collapsed, then the class means need to be maximally separated in order to minimize the cross entropy loss.

Closely related is also recent work [2'], which shows that in the unconstrained feature model loss minimization leads to neural collapse for models trained with cross entropy or with a supervised contrastive loss function. They assumed representations on a sphere, but allowed for weight decay. Similar to this paper, they also experimented on randomly labeled data and with fixed classifier weights. It would be worth adding a discussion about it.

Furthermore, there is [4'], in which deep neural networks are studied using a convex analytic framework. Therein, it is shown that the weight matrices align with the previous layers at optimality. This serves as an explanation of neural collapse (see Corollary 4.3 in the reference). It would be interesting to discuss similarities and differences between [4'] and this work, as convexity is central to both.


### Clarity:

The paper is well structured (global optima result -> local critical points result -> experiments) and easy to read. Good job.
In particular, I like the bullet points after the problem formulation and theorems, which discuss the assumptions of the theorems or point to related work (As written above, the related work sections need more details).\
I also think that it was a good idea to graphically compare the works [24-26] on neural collapse in table 1. (minor remark to the authors: insert some additional white space between the caption and the table to improve readability.)
---

The proofs in the supplementary material are a bit difficult to follow. \
In general, the proof of theorem 3.1 seems to be more involved than the proof of similar results, for example in [2'], [3'].  For the reader it remains unclear, why this is the case and I would appreciate if the authors comment on this in their response to the review.\
In particular, I do not understand, why Lemma D.5 is in the form of "for any $c_1>0$, the cross entropy loss can be lower bounded by...".  As consequence,  all other intermediate results building on lemma D.5 are also of this form, which makes the proofs hard to read.
I might be wrong, but to me it seems, that $c_1$ is introduced artificially, but never really used, as one is only interested in the equality case between lines 945 and 946 anyways, and this does not depend on $c_1$.

### Quality:

The paper includes well designed experiments.\
Experiment 4.1 establishes, that neural collapse occurs independently from the optimization algorithm. Experiment 4.2 studies the unconstrained feature model assumption in greater detail. As one might already suspect, it clearly demonstrates that increasing the model capacity (measured by the width of the convolutional layers) leads to more simplex-ETF like representations. Thus confirming the unconstrained feature model assumption for highly overparametrized networks.
I really liked that the authors introduced metrics (NC1-NC4) for measuring each of the respective neural collapse properties separately.
---

However, I am not convinced, that the experiments demonstrate that the input "plays minimal influence" (line 310) for reaching neural collapse. This is simply not studied in the experiments, as they all operate on the same input data.

---
While I do not doubt the results, the experiments would definitely benefit from more statistical power. So far, only the results of a single run (in each setup) is presented, as acknowledged by the authors in the checklist. However, reporting learning curves does not automatically prohibit to report error bars. For example, one could report mean learning curves and shade the error regions.

---
The theory appears to be correct. However, I have to admit, that I was not able to check all of it thoroughly.

---
The main weakness of the paper is that its fundamental assumption (separating it from other works using the unconstrained feature model) is not justified enough.

The theory relies on the assumption, that the representations are subject to an L2-penalty in the unconstrained feature model. As the authors argue, it seems plausible, that this assumption is  meaningful, i.e. it provides a valid approximation of the representations of a neural network. However, on a closer look, it appears that this assumption is made only so that a connection to a convex optimization problem can be established via Lemma C.3. From this perspective, the validity of the this assumption needs to be discussed in greater details and to be confirmed in experiments.\
For example (other strategies are possible as well), the proof of theorem D.1 could be adapted such that optimal norms of the representations can be predicted (compare Corollary 1 in [2']). Thus, if one knows how the regularization strength $\lambda_H$ depends on the model depth and the weight decay parameter used in practice, this prediction on the norms could then be tested in experiments.

---
The authors state that they make the assumption of balanced class multiplicities in the training set due to simplicity (line 140). However, this is misleading, as this assumption is crucial for the theory (e.g. the proof of Lemma D.3 uses this assumption). If the class multiplicities are unbalanced, the loss minimizer is not a simplex ETF.


### Minor remarks:

the links within the paper do not work

line 289: typo quantity -> quantify

The most left (and maybe also the most right) plots in figures 3 and 4 would probably benefit from a log scale. As of now, the most right plot in figure 4 is unreadable.\
In figure 4 (most left) the values t on the y-axis seems to be labeled incorrectly. What is meant there? t (i.e., as it is now), e^t, 10^t, or something else?\
I might have missed it, but please mention the width of a standard resnet-18 model for comparison.

line 185: "The requirement that d≥K is  necessary  for  Theorem  3.1  to  hold,  simply  because K vectors  in R^d cannot  form  a K-Simplex ETF if K > d."\
I don't think this is true. K vectors can still form a K-Simplex ETF if K=d+1, for example an equilateral triangle in R^2.

**References:** \
[1'] Wang et al., NormFace:l2hypersphere embedding for face verification, ACM Multimedia, 2017\
[2'] Graf et al., Dissecting Supervised Contrastive Learning, ICML, 2021\
[3'] Lu and Steinerberger, Neural collapse with cross-entropy loss, arxiv, 2020\
[4'] Ergen and Pilanci, Revealing the Structure of Deep Neural Networks via Convex Duality, ICML 2021\
[5'] Yun et al., Are deep ResNets provably better than linear predictors?, NeurIPS 2019\
[6'] Hardt and Ma, Identity Matters in Deep Learning, ICLR 2017


**Time Spent Reviewing:**

I don't really know

---

> ### Author Response · Authors · 2021-08-10
> **Response to Reviewer HfZN (Part 2)**
>
> ## Clarifications on the experiments
>
> * **Error bars are not used**. We didn't report error bars because each figure has multiple plots of learning curves which are already overlapping to each other in almost every figure, so adding the error bars will make the plots quite messy. But as we mentioned in the checklist, we did run the experiments multiple times, and observed very similar performance in terms of neural collapse, testing accuracy, and other statistics. We will make this clear in the paper. Nonetheless, we agree with the reviewer that adding error bars is a good idea in general and we will conduct extra experiments and will plot figures with error bars similar to Figure 7 in [2’].
>
> * **Input plays minimal influence for reaching neural collapse**. Although the inputs are the same CIFAR10 dataset, we do consistently observe the same neural collapse from both Figure 4 and Figure 8 (in the Appendix) on completely different random labels with different network architectures. As the overparameterized network is fitting to the labels with neural collapse regardless of the input, we draw the conclusion that the input plays minimal influence for reaching neural collapse.
>
> However, the reviewer has made a very good point on this. During the rebuttal period, we have conducted experiments for random input images with each pixel generated uniformly between 0 and 1,  where we also consistently observe neural collapse. We will add these experimental results and rephrase the sentence to make it more clear in the revision.
>
> ## Other Comments on Writing
>
> Following the reviewer's suggestion, we will address the concern of ''hard to assess its role in the literature'' at a high level, by moving part of the discussion in Appendix A to the main paper and highlight the key references for each topic that we are discussing (including the paper that the reivewer recommended), while maintaining the current citation list. In the following we address some minor comments in detail.
>
> * **Relationship to low-rank matrix recovery**. The connection to low-rank matrix recovery problems is discussed in detail in Section 3.1 (page 6, line 232-239) and Footnote 9. We also refer the reviewer to our response to **Reviewer 7gsu** for more comments as well. In the revision, we will discuss the relationship in more detail.
>
> * **For low-rank matrix recovery: ''I did not find this in section 1, maybe the authors could specify the exact location of the discussion in section 1.''** This refers to the sentence "*As discussed in Section 1, it has been recently shown that the strict saddle property holds for a wide range of nonconvex problems in machine learning [66–79], including low-rank matrix recovery [74, 76, 80–83]*" (line 233).
> Thanks for the good catch. They are discussed in Appendix A (the last paragraph). We will correct this in the revision.
>
> * **Addressing other minor comments**. In the revision, we will make sure to address other minor issues, such as adding other references (like the critical point analysis of residual network) and including more discussions with prior works, inserting additional white space between the caption and Table 1, correcting typos, mentioning the used log scale (which is the natural logarithm), changing some plots to the log scale for better visualization, rephrasing the sentence in line 140 about balanced data, rephrasing the sentences in line 185 to include the case $K = d +1$ for $K$-simplex ETF (Our Theorem 1 also holds for $K = d+1$), and others.
>
> ### References mentioned
> [1'] Wang et al., NormFace:l2hypersphere embedding for face verification, ACM Multimedia, 2017
>
> [2’] Florian Graf, Christoph Hofer, Marc Niethammer, Roland Kwitt, Dissecting Supervised Contrastive Learning, ICML 2021.
>
> [3'] Lu and Steinerberger, Neural collapse with cross-entropy loss, arxiv, 2020
>
> [4'] Ergen and Pilanci, Revealing the Structure of Deep Neural Networks via Convex Duality, ICML 2021
>
> [5'] Yun et al., Are deep ResNets provably better than linear predictors?, NeurIPS 2019
>
> [6'] Hardt and Ma, Identity Matters in Deep Learning, ICLR 2017
>
> [7’] Wenlong Ji, Yiping Lu*, Yiliang Zhang, Zhun Deng, Weijie J Su. How Gradient Descent Separates Data with Neural Collapse: A Layer-Peeled Perspective, 2021.
>
> [8’] Nishanth Dikkala, Gal Kaplun, Rina Panigrahy, For Manifold Learning, Deep Neural Networks can be Locality Sensitive Hash Functions, arXiv preprint arXiv:2103.06875, 2021.
>
> [9’] XY Han, Vardan Papyan, David L Donoho, Neural Collapse Under MSE Loss: Proximity to and Dynamics on the Central Path, arXiv preprint arXiv:2106.02073
>
> [48] Jason D Lee, Max Simchowitz, Michael I Jordan, and Benjamin Recht. Gradient descent only converges to minimizers. In Conference on learning theory, pages 1246–1257. PMLR, 2016.

---

> > ### Comment · Reviewer_HfZN · 2021-08-17
> > **Response to the authors**
> >
> > *Thank you for your deatiled answer. It clears most of my concerns. However, there are some points I would like you to reply to.*
> >
> > Regarding Problem formulation (3) with weight decay on the representations.
> >
> > Thank you for pointing me towards the experiments in Appendix B.2.2. However, they do not really cover my concerns, which are not about whether networks trained with a loss as in Eq. (3) can achieve neural collpase, but about to which extend the idealization from formulation (3) can describe standard neural networks. \
> > This is why I suggested to empirically test the theoretical prediction on the norms of the representations. It seems, that your theory is indeed capable of predicting the norms and I am pleased that you will include results from such experiments in the revision. I want to emphasize, that these experiments should be performed on **network architectures used in practice**, e.g., a ResNet18 as in sections 4.1 and 4.2, and with weight decay on the parameters, but **no weight deacy on the features** . As you already pointed out, your theory already implies that gradient descent finds the global solution if the representaions are freely optimized.
> >
> > Could you also clarify (in a reply and in the paper), how important formualtion (3) is for your theory. In particular, can your proof technique be adapted (and how difficult would it be) to cover different variants of the unconstrained feature model (e.g., without norm constraints or explicit constraints on the $l_2$ norm of representations). I won't hold it again the submission if it cannot, but I would appreciate the honesty.
> >
> > Thank you for the additional discussion of related work.  I am confident that the plan formulated in the "other comments on writing" section in your reply will resolve the issue.
> >
> > Regarding "more involved proof of Theorem 3.1". I think my problems when reading the proof were mostly due to the introduction of the constants $c_1$ and $c_2$, as there role was unclear to me. Thank you for clarifiyng, that this was a notational choice; please include this information in a final version of the paper/appendix.
> >
> > Regarding error bars: I agree, including error regions might make the plots messy and boxplots or histograms are a good idea to visualize the spread. I would recommend to stick to the quantities NC1-NC4 in such plots (as opposed to the quantities reported in Figure 7 in [2’].

---

> > > ### Author Response · Authors · 2021-08-19
> > > **Response to Reviewer HfZN**
> > >
> > > We sincerely thank the reviewer for the appreciation of our responses, as well as more constructive comments. We address them in detail as follows.
> > >
> > > ## Prediction on the norms of the representations for practical networks and explanations on  extend the idealization from formulation (3) can describe standard neural networks
> > >
> > > We sincerely thank the reviewer for further clarifications on this point and constructive comments. As the reviewer correctly pointed out, our theory can predict the norms of the features under the unconstrained feature model and problem formulation (3). In particular, under this setting our theory predicts that ($i$) the norms of all feature vectors are equal, and ($ii$) the value of the norm can be precisely estimated as a function of the $\ell_2$ regularization on the classifier parameters as well as on the features.
> > >
> > > When applied for the practical network (e.g., ResNet18) trained with weight decay on the network parameters, for ($i$), our experiments as well as results in [1] show the representations of practical networks collapse and have the same norm, which is consistent with our theorems; for ($ii$), however, we find it difficult to estimate the norm of the features, because there is no explicit relationship between the $\ell_2$ regularization on the features and that on the parameters of the deep network. This is due to the high nonlinearity of the network.
> > >
> > > Nonetheless, if we want to measure the closeness of our formulation (3) and the practical weight decay formulation, we think the closeness of the neural collapse and the closeness of the generalization performance might be good metrics. Our results in Appendix B.2.2 showed that both of them are close.
> > >
> > > ## Clarification on the importance of problem formulation (3) and extension to other variants
> > >
> > > Under the constraint feature model, we believe the benign global geometry of the optimization landscape is induced by the intrinsic rotational symmetry of the problem, so that it is *not* depending on the particular formulation such as (3). Thus, we believe the benign global landscape properties can be extended to other constrained variants of our formulation. For example, recent work [7’] showed a similar (but a bit weaker) landscape result under a different problem formulation.
> > >
> > > As pointed out by **Reviewer 7gsu**, one natural idea to extend our results to the constrained formulation (e.g., constraints of $\ell_2$-norm on the features and classifiers) might be through the Lagrangian function and Lagrange duality, which relates the two formulations. More specifically, based on the strict saddle property for our problem formulation (3), we may directly construct the negative curvature direction for the saddle points of the constrained formulations by using Lagrange duality between the constrained and unconstrained ones. Indeed, this could be a good extension of our work that we will discuss in the revised version. We leave the formal justification as future work.
> > >
> > > [1] Vardan Papyan, XY Han, and David L Donoho. Prevalence of neural collapse during the terminal phase of deep learning training. Proceedings of the National Academy of Sciences, 117(40):24652–24663, 2020.
> > >
> > > [7’] Wenlong Ji, Yiping Lu, Yiliang Zhang, Zhun Deng, Weijie J Su. How Gradient Descent Separates Data with Neural Collapse: A Layer-Peeled Perspective, 2021.

---

> > > > ### Comment · Reviewer_HfZN · 2021-08-20
> > > > **Response to the authors**
> > > >
> > > > Thank you for the further comments.
> > > >
> > > > It seems that we cannot reach a final conclusion on the validity of assumption (3), i.e. how good of a model for a real neural network it is. Importantly however, it is able to imply the empirically observed neural collapse phenomenon.
> > > >
> > > > I am quite positive about the paper now that I got your responses and I raise my score to accept.

---

> > > > > ### Author Response · Authors · 2021-08-20
> > > > > **Response to Reviewer HfZN**
> > > > >
> > > > > We thank the reviewer for the appreciation of our efforts in the response and we will incorporate these discussions in the revision.

---

> ### Author Response · Authors · 2021-08-10
> **Response to Reviewer HfZN (Part 1)**
>
> We would like to thank the reviewer for their detailed and enlightening review, helping us increase the quality of our work. First of all, as the study of neural collapse and deep representation learning, in general, is a very rapidly developing area, several very related papers to our result have been released just before/after the submission of our paper (such as [2’,4’,7’-9’]), which were missed for our submission. We will cite these papers in the revision, and discuss the relationship in detail. In the following, we address the reviewer’s comments one by one.
>
> ## Problem formulation (3) with weight decay on the representations.
>
> In the current practice of deep learning for classification problems, one of the most commonly used loss functions is cross-entropy with weight decay (i.e., $\ell^2$-regularization) on the network parameters $\bf \Theta$, implying that the norm of the features $\bf H$ is also implicitly penalized. The above observation naturally motivates us to consider enforcing the $\ell^2$-regularization on the feature $\bf H$, which, under the unconstrained feature model,  is the closest formulation to the above practically used loss (in comparison to other constrained formulations listed in Table 1). Admittedly, as the reviewer has pointed out, there are subtle differences between the two losses. We address the reviewer’s concerns as follows.
>
> * **Experimental verification**. In Appendix B.2.2, we have conducted extra experiments comparing regularizations on all network parameters $\bf \Theta$ and regularizations on $\bf H$ to justify our problem formulation (3). In particular, among lines 803-807 in Appendix B.2.2, we showed that training on both formulations reaches comparable generalization performance on CIFAR and MINST datasets with ResNet18. Additionally, Figure 9 showed that training with regularizations on $\bf H$ presented a  more pronounced neural collapse than with regularizations on the parameters $\bf \Theta$. Thus, the experimental results imply that our formulation (3) is not only of theoretical interest but also of practical use.
>
> * **Prediction of the norm of the optimal representations**. Under the constrained feature model, the norm of the optimal representations can also be predicted through the proof of Theorem D.1. In the revision, we will add a similar figure as Figure 4 in [2’]. It should be noted that Figure 11 in Appendix D.1 gives the minimal function value with respect to the norm of the optimal representations, in a similar flavor to Figure 4 in [2’]. In comparison, we have not plotted the empirical dots as in Figure 4 in [2’]. This is because our global geometric result in Theorem 3.2 already implies gradient descent almost surely finds the global solution, thus matching the theoretical results.
>
> In the paper, we have briefly discussed the validity of our loss in Section 2.1 (page 4, line 151 - 157) and experimentally verified in Appendix B.2.2. Following the reviewer’s suggestions, we will make this more clear in the revision.
>
>
>
> ## Discussing the relationships to other related work
>
> As pointed out by Reviewer xttS, we may have cited more papers than usual because we want to acknowledge everyone’s contribution around this topic and give credit to others that inspired us, making the manuscript more valuable to other researchers as well. The reason we only discussed superficially for most references is due to space limitations for a conference paper. However, the reviewer’s point is well taken, and following these suggestions, we will discuss in more detail several key related references (especially the ones pointed out by the reviewer), that we elaborate on in more detail below.
>
> * **Overall comparison in terms of results**. We very much appreciate the reviewer for timely pointing us to some recent related work [1’,2’,4’] that we were unaware of upon the submission. In the revision, we will cite [1’,2’,4’] and compare them with our work in Table 1. As pointed out by the reviewer, [1’] studied equinorm weights and already collapsed classes, which is a special case of ours. The work [2’] constrains the features inside a ball for the cross-entropy loss and characterizes the global optimality of the cross-entropy loss. The work [4’] considers the least-squares loss and shows that the solutions with neural collapse are optimal solutions, but it does not exclude the existence of other global solutions or local solutions. In contrast, our result not only characterizes global optimality (Theorem 3.1), but also the global optimization landscape (Theorem 3.2): for the unconstrained features, it shows that there is no spurious local minimum, and any other critical point that is not a global minimum is a strict saddle, ensuring convergence to the global minimum for common iterative algorithms like gradient descent [48]. As summarized in Table 1, the landscape analysis is one of the main contributions of our work compared with other related work.
>
> * **Response to "more involved proof of Theorem 3.1 than [2’,3’]" & clarification of Lemma D.5**. As mentioned above, [2’] considers the features inside a $\ell_2$-norm ball for the cross-entropy loss (and then the $\ell_2$-regularization and an $\ell_2$-norm ball constraint on the classifiers), while we add weight decay on both the features and linear classifiers. Thus, we need additional efforts to balance the energies of the features and classifiers (as in Lemma D.2), making sure that both of them are finite (as in Figure 11). Admittedly, we notice that there are common proving techniques used in both [2’] and ours. The work [3'] assumes collapsed features on the unit sphere and analyzes contrastive loss operating directly on the features without the linear classifier. Because our formulation involves both the linear classifier and features as optimization variables, our proof is more involved than that in [3’]. With the caveat in mind, we think it is possible to simplify our proof and will make it more concise in the revision.
>
> For Lemma D.5, the reason we organized our proofs in this way actually is to make them more readable. Lemma D.5 presents a lower bound for the cross-entropy in the simplest form of $\bf z$, then we build upon it for other more specific intermediate proofs. Thus, the readers can still understand the main proof and other lemmas by assuming Lemma D.5 is true, without going into every technicality of Lemma D.5. If this is counterproductive, we will revise in the revision.
>
> * **Comparison with [4’] in terms of proving techniques**. While the proofs in [4’] and our work both involve the connection to convex problems, they serve substantially different purposes. In [4’], convexity is involved in proving strong duality for the solutions that have neural collapse, thus certifying them as optimal. In contrast, we use the connection to the convex problem to construct negative curvature directions for those critical points that are not global minima of a nonconvex loss.
>
> ### to be continued in the following

---

### Official Review · Reviewer_Rfbc · 2021-07-17

**Rating:** 8
**Confidence:** 4

**Summary:**

Neural collapse is an intriguing phenomenon that has been observed in neural network classifiers as the training error goes to 0 where the last layer of the neural network has collapsed input points to corners of a simplex, thereby minimizing variability, and maximizing separability. While this phenomenon seems well studied the authors extend it by studying the critical points of this optimization geometrically by relating it to a convex problem. They show in Thrm 3.2 that the critical points are either global minimal or saddle points with negative curvature which can be escaped by a minimizer.


**Limitations And Societal Impact:**

Yes

**Main Review:**

This work builds incremental, but important advances on what seems to be well-investigated theory regarding deep neural networks trained for K-class classification via a cross entropy loss, neural collapse. Neural Collapse was first reported in [1, 16] Key parts of the theory include the unconstrained feature model which involves analyzing only the last layer , and relating the non-convex optimization to a convex problem to characterize the minima. These parts of the theory have all been introduced and used in various works such as [23] and [30] respectively. To that this work extends these ideas a bit. They show that using [30] one can characterize the critical points as having at least one direction of negative curvature, by constructing a negative curvature direction. They then show that this that sufficiently powerful neural networks can escape these saddle points and reach a global optimum, and that this is independent of the optimization algorithm. They also propose an architectural choice of d=K in the last dimension, which is used frequently regardless. A more interesting idea is the fixing of the classifier.

The writing of the paper however seems list-like and with a piece of theory and minor ideas for practice. Instead I would consider first explaining the previous theory fully, particularly the unconstrained feature model, which as the authors say in the supplement may be an alternative condition to the "infinitely wide condition of the NTK" for casting the neural network as a linear model in an alternative feature space. What is the intuition behind the unconstrained feature model? Does it imply that those layers are not necessary or that they can be trained randomly? Some experiments here would help as well, changing the width for example of previous layers.

Second, the authors need to provide the background of [30] to address their geometric idea, which is in the title but not covered in the main paper. I believe this is the main contribution of the paper. This is what makes this idea practical and apply to modern DL training methods. Perhaps an additional metric would be to actually estimate the curvature of the landscape using Olivier-Ricci curvature or other data estimators of curvature would emphasize this points. See: https://www.sciencedirect.com/science/article/pii/S1631073X07004414?via%3Dihub.

Finally, I think that d=K is somewhat obvious as an architectural choice, particularly given one hot encodings commonly used in classifications. I would rather understand more about fixing the last layer. Does the exact simplex used matter? How is it enforced?

**Time Spent Reviewing:**

5 hours

---

> ### Author Response · Authors · 2021-08-10
> **Response to Reviewer Rfbc**
>
> We thank the reviewer for the detailed and thoughtful comments. In the following, we address the reviewer’s comments in detail.
>
> ## Clarifications on the unconstrained feature model.
>
> 1. **Intuition and motivation.** Our motivation for studying neural collapse under the unconstrained feature model is due to the following facts:
> 	* Nonlinear interaction across many different (shallow) layers creates tremendous difficulty in the analysis of optimization, while the characterization of neural collapse only involves the last-layer features and classifiers.
> 	* Modern deep neural networks are often highly overparameterized with the capacity of learning \*any* representations, so that the last-layer features can approximate, or interpolate, any point in the feature space (see the references [41,42,43,44]). The unconstrained feature model most directly follows this observation.
> In our current manuscript, we have discussed the above facts in a bit more detail in Section 1 (page 2, line 48 - 61) and Section 2 (page 5, line 158 - 167), and we will expand the discussion in more detail in the revision.
>
> 2. **Necessity of previous layers.** Our result does NOT imply that the previous layers are not necessary nor that they can be trained randomly. Indeed, as mentioned before, the unconstrained model relies on the ability of the network to learn representations for the input data. Also, note that the choices of network architectures and training methods are critical for good generalization and robustness, due to the fact that the generalization and robustness performance of deep network critically depend on the properties such as Lipschitzness constants of the network, which cannot be captured by the unconstrained feature model. Following the reviewer's suggestion, we will make these points clearer in the revision.
>
> 3. **Experimental validation**. In our paper, we have studied the effects of previous layers and demonstrated the validity of the unconstrained feature model by experimental results in Section 4.2 and Appendix B.2. In particular,
> 	* **Changing the Width of Previous Layers**. In Section 4.2, we trained the network with different widths across layers on ResNet18 and CIFAR10, where Figure 4 showed that the training exhibits more severe neural collapse when the network is wider (i.e., more overparameterized). This corroborates with our assumption that a highly overparameterized network can almost approximate, or interpolate, any point in the feature space;
> 	* **Different training algorithms have different implicit biases**. As in Appendix B.2.1 and Figure 7, although different training algorithms exhibit very similar neural collapse, they learn different weights with different implicit biases, leading to different generalization performance. This implies that the choices of the weights are important in terms of generalization.
>
> Following the reviewer's suggestion, we will move Figure 4 earlier to Section 2.1 in the revision and move Figure 7 to the main paper.
>
> ## Comparison with the work [30]
>
> There are crucial differences between the work [30] and our results. The work [30] only characterizes the global optimality conditions (indeed a sufficient condition for a local minimum to be optimal), which neglects several important aspects of training deep networks: ($i$) the geometric properties of critical points other than global solutions, ($ii$) the kind of representations that are learned at global solutions, and ($ii$i) how efficient algorithms achieve these global solutions with theoretical guarantees. In comparison, our work addresses the above questions under the unconstrained feature model. Through a finer geometric analysis of all critical points, we showed ($i$) the neural collapse solutions are the only global solutions up to rotations, and ($ii$) all other critical points are strict saddle points with benign geometric properties that can be escaped using negative curvature (i.e., the Hessian has negative eigenvalue). This finding is similar to recent advances on other nonconvex problems [47]. Our geometric result ensures that local search algorithms, such as gradient descent, can find the global minimum [48]. Though the negative curvature of the landscape can be exploited for algorithm design, common algorithms like gradient descent can almost surely escape the strict saddles without explicitly using the negative curvature [48].
>
> We will incorporate the above discussion in the revision.
>
> ## The choice of $d=K$ is somewhat obvious as an architectural choice
> While the design option of employing d=K follows naturally from our theoretical analysis, we want to emphasize that this is neither a commonly used nor an obvious architectural choice in deep learning models. Recall that $d$ refers to the dimension of the features, while $K$ is the number of classes. Popular ResNet models employ $d=512$ for CIFAR10 (where $K=10$), or $d=2048$ for ImageNet (where $K=1000$). We will highlight this in the revision.
>
> ## Choices of Simplex ETF used in the last layer
> For experiments in Section 4.3, the Simplex ETF is chosen to be the canonical one, e.g., when $K = d$, ${\bf W} = {\bf I_K} - 1/K {\bf 1}_K{\bf 1}_K^\top$, as stated in Footnote 12. However, the choices of specific Simplex ETFs have negligible effect: in theory, we showed that the simplex ETFs are global solutions up to arbitrary rotations in Theorem 3.1 and Eq. (5); in experiments, we observe very similar performance when the simplex ETF is randomly rotated. We will add this discussion and move footnote 12 to the main text, following the reviewer’s suggestions.
>
> ### References mentioned
> [30] Benjamin D Haeffele and Rene Vidal. Global optimality in tensor factorization, deep learning, and beyond. arXiv preprint arXiv:1506.07540, 2015.
>
> [47] Yuqian Zhang, Qing Qu, and John Wright. From symmetry to geometry: Tractable nonconvex problems. arXiv preprint arXiv:2007.06753, 2020.
>
> [48] Jason D Lee, Max Simchowitz, Michael I Jordan, and Benjamin Recht. Gradient descent only converges to minimizers. In Conference on learning theory, pages 1246–1257. PMLR, 2016.

---

### Official Review · Reviewer_xttS · 2021-07-27

**Rating:** 8
**Confidence:** 5

**Summary:**

This is an extremely well written (articulate and readable) and diligently prepared (completely
by the book) manuscript. For example, it draws carefully on more than 100 other manuscripts.
This alone makes the manuscript valuable to other people, as it teaches how work can
be exposited and papers can be written.

The results are interesting and instructive, and in some way connect
these questions to some recent work in non-convex optimization.
The practical benefit of this work -- tens of percentage points speedups --
are meaningful.



**Limitations And Societal Impact:**

In a few cases the paper betrays its authors' obvious theoretical
origin. Theorists still believe that deep learning is somehow
in a problematic situation because of its reliance on empiricism and
phenomenology and the "mysteries" unearthed by empirical science.
This is simply a parochial viewpoint. It would be better for theorists to recognize that
the state of the art in deep learning is an empirical triumph and that
it's better for theory to contribute to systematizing the various lessons that
are being discovered by active machine learning researchers.

**Main Review:**

Originality: yes this is original.
Quality: this is very well prepared as an academic work.
Clarity: this is exceptionally clear.
Significance: this paper shows that in this field,
theory can eventually make sense
of phenomena which have been discovered
and carefully documented empirically. The
message of science progressing in this way is
worth spreading far and wide. Generally speaking theorists
working in this area have not been concerned with explaining empirically
uncovered phenomena. So this example may lead to a new
synergy.

**Time Spent Reviewing:**

2

---

> ### Author Response · Authors · 2021-08-10
> **Response to Reviewer xttS**
>
> We very much thank the reviewer for the appreciation and support of our work. We also appreciate the reviewer’s critical perspective on the current status of deep learning theory and practice. It is also our interest and goal to systematize the various lessons and mathematical principles behind many intriguing phenomena that have been empirically discovered in deep learning.
>
> Meanwhile, since the study of neural collapse and deep representation learning, in general, is a rapidly developing area, several very related papers to our result have been released just before/since the submission of our paper, for example,
>
> [1’] XY Han, Vardan Papyan, David L Donoho, Neural Collapse Under MSE Loss: Proximity to and Dynamics on the Central Path, arXiv preprint arXiv:2106.02073
>
> [2’] Florian Graf, Christoph Hofer, Marc Niethammer, Roland Kwitt, Dissecting Supervised Contrastive Learning, ICML 2021.
>
> [3’] Wenlong Ji, Yiping Lu, Yiliang Zhang, Zhun Deng, Weijie J Su. How Gradient Descent Separates Data with Neural Collapse: A Layer-Peeled Perspective.
>
> [4’] Nishanth Dikkala, Gal Kaplun, Rina Panigrahy, For Manifold Learning, Deep Neural Networks can be Locality Sensitive Hash Functions, arXiv preprint arXiv:2103.06875
>
> We will further cite these papers in the updated version, and discuss the relationship in detail.

---

### Decision · Program_Chairs · 2021-09-27

**Decision:**

Accept (Spotlight)

**Comment:**

This paper provides an interesting well presented study of neural collapse, mostly focusing on solid theoretical results, but also demonstrating their empirical implication. All reviewers agree the paper should be accepted, commending the high quality of the presentation in the paper. Further, it is clear from the reviews and the paper that this is an important problem, and the results are sufficiently significant to be of interest to a wide audience at NeurIPS (as evidence, two reviewers marked the paper as top 50% of accepted NeurIPS papers with high or absolute confidence). Therefore, I recommend acceptance as a spotlight.

To the authors: please read carefully the reviewer comments and follow up on your responses with appropriate revisions to the paper.